# The Role of Vegetation Monitoring in the Conservation of Coastal Habitats N2000: A Case Study of a Wetland Area in Southeast Sicily (Italy)

**Saverio Sciandrello [1,*]**, **Veronica Ranno [1]** and **Valeria Tomaselli [2]**

[1] Department of Biological, Geological and Environmental Sciences, University of Catania, Via A. Longo 19, 95125 Catania, Italy; veronica.ranno@phd.unict.it

[2] Department of Biosciences Biotechnologies and Environment, University of Bari "Aldo Moro", Via Orabona 4, 70125 Bari, Italy; valeria.tomaselli@uniba.it

\* Correspondence: s.sciandrello@unict.it

**Abstract:** The coastal wetlands are among the most vulnerable and threatened environments of the Mediterranean area. Targeted actions for their conservation require in-depth knowledge of current and past natural vegetation. In this paper, we surveyed the vegetation composition and the spatio-temporal changes of a coastal wetland area in southeastern Sicily ("Saline di Priolo" SAC). Based on 128 phytosociological surveys and several plant collections, a total of 304 taxa, 28 plant communities, and 16 habitats have been identified. Furthermore, three new plant associations were described, including two in wetland and one in rocky coast environments. For the classification of plant communities and habitats, a hierarchical clustering was performed by using Euclidean coefficient and beta-flexible algorithm. The life form spectrum of the current flora highlights the dominance of therophytes and hemicryptophytes. The Mediterranean species are largely prevailing with 123 taxa. The cartographic analysis performed with ArcGis 10.3 shows a radical reduction in the wetland habitats in the last 70 years, and a strong alteration of the ecological succession of the psammophilous-hygrophilous vegetation. Moreover, landscape configuration of the coastal vegetation and habitat types was well highlighted by a set of specific landscape metrics. In particular, our outcomes identify three habitats (2110, 2210, and 5220\* EU code) with bad conservation status, among which we identified one of priority conservation (*Zyziphus* arborescent matorral) that requires urgent restoration measures.

**Keywords:** conservation; diachronic analysis; landscape fragmentation; plant diversity

## 1. Introduction

Mediterranean wetlands and their adjacent areas are extremely important for biodiversity and human wellbeing, but their integrity has often been compromised [1]. Fortunately, in recent times, opinion movements and policy actions aimed at wetland recovery and protection have arisen [2]. Wetlands are essential ecosystems for preserving animal and plant diversity in the Mediterranean region. These biotopes host highly specialized plant communities linked to the substrate type, water quality, flooding time, and bioclimate, in which rare species grow with specific ecophysiological adaptations [3]. These habitats are disappearing due to human pressure, especially when these regions are subject to land reclamation [4]. This is the case of southern Italy, Sicily, and nearby islands, where numerous coastal wetlands have been subjected to severe degradation in recent decades [5,6]. In Sicily, coastal wetlands are located mainly in the west, between Trapani and Mazara del Vallo and in the southeastern sectors of the island, and along the coastal strip between Siracusa and Catania. The knowledge about the coastal saltmarsh vegetation in Sicily was investigated by several phytosociological studies [5,7–12]. In southeastern Sicily, the Special Area of Conservation (SAC) "Saline di Priolo" (ITA090013), despite several environmental

changes occurring in the last 70 years, is distinguished by a significant diversity of habitats and plant species. This area includes the Natural Reserve "Saline di Priolo" and comprises different environments, including beaches, cliffs, brackish marshes, grasslands, temporary ponds, etc. The SAC "Saline di Priolo", in addition to an extensive system of coastal wetlands and saltmarshes, also includes the "Magnisi" peninsula, and the archaeological sites known as Thapsos and "Biggemi" lagoon. This area is under considerable anthropogenic pressure from nearby industrial activity; in fact, immediately next to the protected area is the Syracuse petrochemical hub. It is a vast industrialized area (the largest Italian petrochemical hub, and the second in Europe) created in the 1950s which extends over an area of 40 km$^2$ on the east coast of Sicily, including the territory of the municipalities of Augusta, Priolo Gargallo, and Melilli up to the gates of Syracuse, and exerting a dramatic environmental pressure on the surrounding territories [13]. As regards the flora and vegetation of the Priolo wetland, only a few significant investigations have been carried out [9,12,14–16]. This study is aimed at providing a comprehensive and updated list of the flora and habitats present in the area, also including species of conservational interest and alien species, with new insights for future management activities of the territory. A diachronic approach, using mapping and monitoring of the flora, vegetation, and habitat of the Nature Reserve and SAC "Saline di Priolo" over the last 70 years, has been carried out, as part of the "MEDISWET" project of the MAVA foundation (Action Activity 7.1.18). Moreover, quantifying the landscape's spatial structure provides a better understanding of the ongoing impact on ecological processes. In this framework, landscape metrics (LMs) are an essential tool to analyze the spatial arrangement of the landscape structure over time [17,18]. The application of LM is particularly suitable to coastal areas because these landscapes are prone to rapid transformations [19].

## 2. Materials and Methods

### 2.1. Study Area

The study area is located on the coastal side of the Hyblean plateau (southeastern Sicily), which is one of the northernmost promontories of the African plate. The Hyblaean plateau (or Hyblaean–Maltese Plateau) is a crust of the continental type isostatically raised and well-defined on its edges, different from the other ones in Sicily, and extending south to the Maltese Islands, from which it is separated by a continental shelf [20]. From a floristical point of view, the Hyblean territory hosts a lot of peculiar endemic and rare species, such as *Urtica rupestris*, *Zelkova sicula*, *Trachelium lanceolatum*, *Anthemis pignattiorum*, *Limonium pachynense*, *Romulea variicolor*, *Elatine macropoda*, and *Solenopsis laurentia* subsp. *hyblaea* [21,22]. Moreover, it is an area extremely rich in terms of vegetation, hosting a great variety of vegetation types, as well as numerous habitat types according to the Annex I of the 92/43 EEC Directive (or Habitat Directive), including several priority habitat types such as 3170* "Mediterranean Temporary Ponds". The "Saline di Priolo" SAC extends over 252.5 ha and its perimeter falls within the Priolo Gargallo municipal area. Average annual rainfall is around 500 mm and average annual temperatures are around 18 °C [23]. From a bioclimatic point of view, the area under investigation falls within the thermo-Mediterranean zone, with a dry ombrotype [24].

### 2.2. Phytosociological Data and Taxonomy

The field surveys were carried out mostly from spring 2021 to autumn 2022, according to the Braun–Blanquet phytosociological approach [25]; these relevés were integrated with other data previously collected in 2008, thus achieving a dataset consisting in a total of 128 relevés. Specimens were identified following the second edition of the Flora d'Italia [26–29], while the nomenclature of vascular plants follows Portal of the Flora of Italy and subsequent updates [30,31], and the regional distribution "rarity" follows Giardina et al. [32].

*2.3. Statistical Analysis and Classification*

For vegetation classification and identification of plant communities, a hierarchical clustering was performed in the program PC-ORD 6 [33], by using Euclidean coefficient and beta-flexible algorithm, according to [34]. Syntaxonomic classification follows Biondi et al. [35] and Mucina et al. [36]. Natural vegetational types were classified into habitat types according to the Italian interpretation manual of habitats [37].

*2.4. Landscape and Diachronic Analysis*

To evaluate the conservation status of the vegetation and habitat types throughout the last 70 years, a diachronic analysis using aerial orthophotos taken in 1955 and 2015 has been carried out, with the production of vegetation and habitat maps achieved with ArcGis 10.3 software. Finally, in order to analyze and evaluate vegetation mosaics and changes in spatial arrangement of the landscape structure over the considered timespan, a set of landscape metrics (LMs) was selected and applied to the two habitat maps (1955 and 2015). Considering the differences in spatial and thematic resolution between the two years of observation, implying a certain level of uncertainty in identifying the single plant associations in the 1955 orthophotos, the landscape metrics were applied to the habitat maps, which have a thematic resolution less detailed than the phytosociological vegetation map. In selecting the most appropriate LM for the fragmentation assessment in coastal Mediterranean environments, we referred to Cushman et al. [38] and to Tomaselli et al. [19]; a limited set of metrics was selected, specifically CA (Class Area), NumP (Number of Patches), MPS (Mean Patch Size), MPAR (Mean Perimeter–Area Ratio), MSI (Mean Shape Index), and AWMSI (Area-Weighted Mean Shape Index), and the analysis was carried out by using the Patch-Analyst extension of ArcGis.

## 3. Results and Discussions

*3.1. The Vascular Flora: Trait Analysis*

A total of 304 taxa were recorded, of which there were 196 for the "Saline di Priolo" and 178 for the Magnisi peninsula (Floristic Appendix A). Overall, 59 families were recorded, with the most represented being *Asteraceae* with 15% (45 sp.), followed by *Poaceae* with 13% (39 sp.) and *Fabaceae* with 12% (38 sp.). The life forms detected show the typical Mediterranean pattern. In fact, the therophytes, with 123 species, represent 40% of the whole flora, followed by the hemicryptophytes with 31% (94 sp.). The percentage of geophytes is also considerable, with 13% (39 sp.), followed by phanerophytes (16 sp.) with 6%, nanophanerophytes (6 sp.) with 2%, and hydrophytes (4 sp.) with 1%. The chorological analysis shows the clear prevalence of the *Mediterranean* chorotype, which represents 42% of the taxa (129 sp.), whereas the *Euro-Mediterranean* (62 sp.) and *Cosmopolitan* (33 sp.) make up, respectively, 20% and 11% of the flora. The *Mediterranean-Turanian* and *Paleotemperate* chorotypes follow (each with 14 sp.) with 15%, while the endemic species (6 sp.) constitute the 2%, which is still relevant considering the limited size of the study area.

*3.2. Taxa of Relevant Interest*

Field surveys highlighted some rare or endemic species, such as *Limonium syracusanum*, *Ziziphus lotus*, *Poterium spinosum*, *Bulliarda vaillantii*, *Damasonium bourgaei* (*Magnisi peninsula*), *Teucrium scordium* subsp. *scordioides*, *Limonium narbonense*, *Euphorbia hirsuta*, and *Cressa cretica* ("Saline di Priolo") (Figure 1).

The most interesting ones are briefly commented on in the following paragraphs.

*Limonium syracusanum* Brullo (Plumbaginaceae).

Suffruticose chamaephyte endemic to the Hyblean Ionian coast, between Augusta and Capo Passero (south-eastern Sicily) [39]. The species characterizes the halophilous phytocoenoses of the rocky coasts of the *Crithmo-Limonietea* class, together with *Crithmum maritimum* and *Arthrocaulon meridionale*. It is included in the red list of Italian flora [40] with the status of least concern (LC). The species has been observed along the reefs of the north and northeastern sector of the Magnisi peninsula, near the lighthouse.

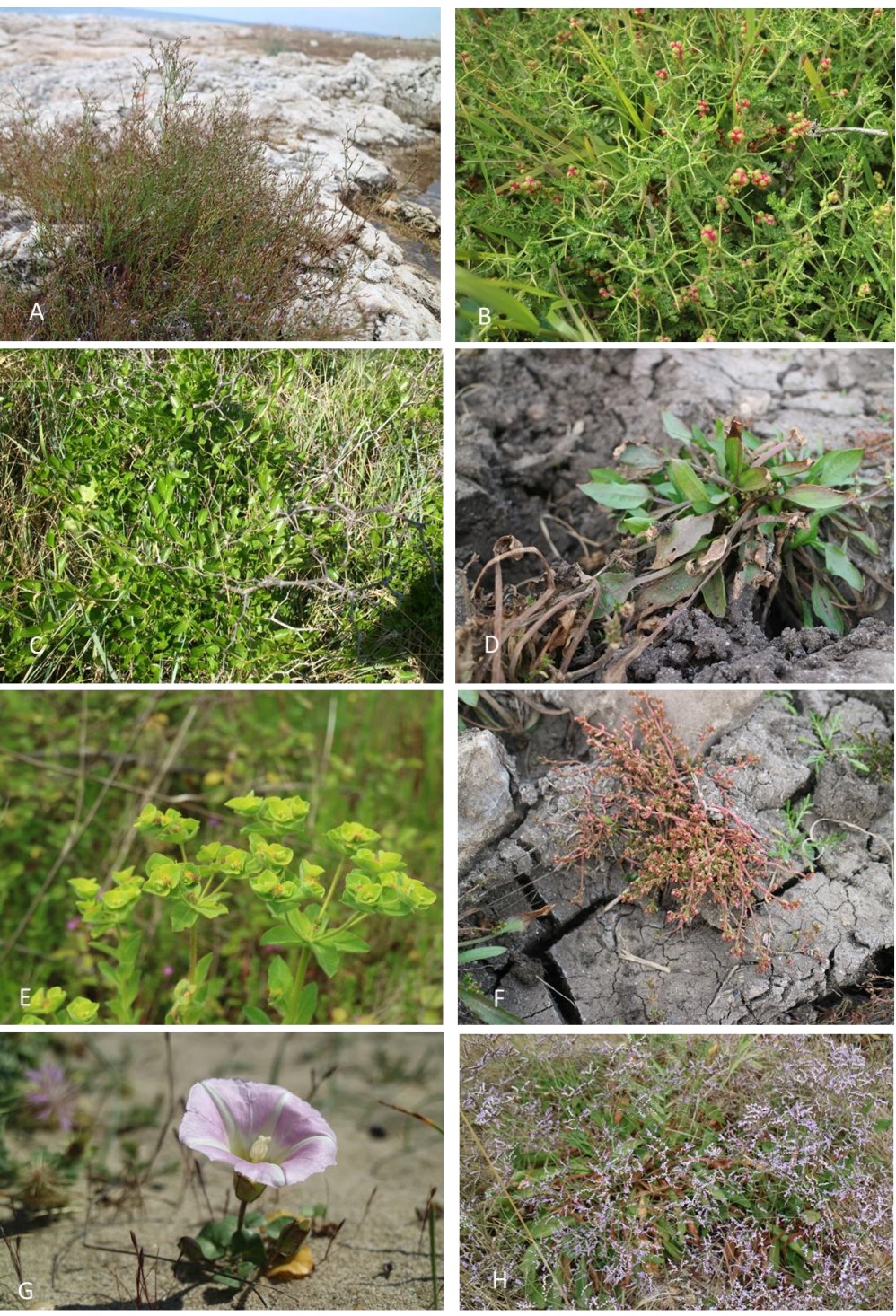

**Figure 1.** Photo plate illustration of some species of the "Saline di Priolo" SAC: (**A**) *Limonium syracusanum*; (**B**) *Poterium spinosum*; (**C**) *Ziziphus lotus*; (**D**) *Damasonium bourgaei*; (**E**) *Euphorbia hirsuta*; (**F**) *Bulliarda vaillantii*; (**G**) *Convolvulus soldanella*; (**H**) *Limonium narbonense*.

*Ziziphus lotus* (L.) Lam. (Rhamnaceae).

Xerophilous deciduous shrub species with a southern Mediterranean–Saharan range. Reported in the red list of Italian flora [40] with the status of least concern (LC). In Sicily, the species is rather rare and, in addition to the studied site, it was reported only for the western sector in M. Pellegrino and Mazara del Vallo [32]. Recently, La Mantia and Scuderi [16] gave a more detailed distribution of the species, confirming the location close to

"Saline di Priolo". At the regional level, Conti et al. [41] reported this species as vulnerable, while Orsenigo et al. [42] listed it as near threatened (NT). In the Magnisi Peninsula, the species was observed for the first time by Zodda [43]; at present, it is represented by a few individuals, limited to a small area on the limestone plateau.

*Poterium spinosum* L. (Rosaceae).

Thorny chamaephyte with East Mediterranean distribution. In Italy, the species is present in Apulia, Basilicata, Calabria, Sicily, and Sardinia. In Sicily, the species is localized exclusively in the Hyblaean area, preferring both carbonate and volcanic substrates [44]. Gargano et al. [45] classified the species as endangered at the national level (EN). Recently, Orsenigo et al. [42] confirmed the status as endangered (EN) for the Italian territory. In the study area, the species was sporadically observed along the coast, in the southern part of the peninsula, and in the garrigues dominated by *Thymbra capitata* and *Micromeria graeca*.

*Bulliarda vaillantii* (Wild.) DC. (Crassulaceae).

Small therophyte with Mediterranean–tropical distribution. It is an amphibious species typical of Mediterranean temporary ponds (Habitat 3170*: Mediterranean temporary ponds). It is a characteristic species of the annual amphibious communities referred to as *Lythro hyssopifoliae-Crassuletum vaillantii* of the *Isoeto-Nanojuncetea* class. In Italy it is present in Sicily, Apulia, Basilicata, Lazio, Tuscany, Liguria, and Sardinia. In the study area, the species is localized in the small temporary ponds in the central part of the Magnisi peninsula.

*Damasonium bourgaei* Coss. (Alismataceae).

Rooting hydrophyte with an Atlantic–Mediterranean distribution. Annual species typical of Mediterranean temporary ponds, growing on carbonate and volcanic substrate. It characterizes the annual amphibious communities of the *Isoeto-Nanojuncetea* class (Habitat 3170*: Mediterranean temporary ponds). In Italy it is present in Apulia, Basilicata, Sardinia, and Sicily. *D. bourgaei* was reported for many coastal sites of Sicily, but at present, it can be considered quite rare. In the Magnisi peninsula, the species is rather localized.

*Teucrium scordium* L. subsp. *scordioides* (Schreb.) Arcang. (Lamiaceae).

Scapose hemicryptophyte with a Euro-Caucasian to NW African distribution range. It typically grows in wet meadows, where it forms hygrophilous communities together with *Juncus subulatus*, *Lotus corniculatus* subsp. *preslii*, *Phyla nodiflora*, and *Potentilla reptans*. In Sicily, it is threatened by the reduction in its natural habitat [32]. In the study area "Saline di Priolo", it is very localized.

*Limonium narbonense* Mill. (Plumbaginaceae).

Rosulated hemicryptophyte with Euro-Mediterranean distribution. Perennial halophilous species typical of the coastal saltmarshes. This species, together with other halophytes, such as *Arthrocaulon meridionale*, *Salicornia perennis* subsp. *alpini*, and *Halimione portulacoides*, characterizes the perennial halophilous vegetation belonging to *Salicornietea fruticosa* class. In the "Saline di Priolo", *L. narbonense* is quite common.

*Euphorbia hirsuta* L. (Euphorbiaceae).

Rhizomatous geophyte with Mediterranean–Macaronesian distribution. The species mainly grows in wet meadows. In the Saline di Priolo, it thrives together with *Teucrium scordium* subsp. *scordioides*, *Lotus corniculatus* subsp. *preslii*, *Phyla nodiflora*, and *Potentilla reptans*. In the Saline di Priolo, the species is very localized, on the edges of small wet areas in contact with *Tamarix africana*. In the last 50 years, the species has undergone a strong reduction in its range along the Sicilian coast; in fact, today there are few areas where the species is conserved [32].

*Cressa cretica* L. (Convolvulaceae).

It is a thermo-cosmopolitan halophilous species growing in sandy or muddy saline habitats. In the study area, the species is rare due to reduction in or alteration of its natural habitat (1310 "*Salicornia* and other annuals colonizing mud and sand"). *Cressa cretica*, together with other annual succulent plants with a summer cycle, characterizes the *Cressetum creticae*, halo-subnitrophilous and termophilous vegetation colonizing abandoned

fields after farming on clayey and salty soils [5]. According to Oresenigo et al. [42], it is classified as "endangered" (EN) in Italy.

### 3.3. Plant Community Description

A total of 28 plant communities (seven annuals and twenty-one perennials) belonging to 18 phytosociological classes has been identified by cluster analysis (Appendix B).

### 3.3.1. Coastal Dune Vegetation (Figure 2; Table S1)

The psammophilous vegetation immediately adjacent the aphytoic belt is characterized by annual herbaceous communities ascribable to the association *Salsolo-Cakiletum maritimae* (habitat 1210 "Annual vegetation of drift lines"). On the embryonic dunes, it is possible to distinguish two different associations. The first one is the perennial herbaceous vegetation of the *Cypero capitati-Agropyretum juncei*, with the dominance of *Thinopyrum junceum* (habitat 2120 "Embryonic shifting dunes"); the second one is the *Sileno coloratae-Ononidetum variegatae*, annual vegetation characterized in particular by *Ononis variegata* and *Silene niceensis* (habitat 2230 "Malcolmietalia dune grasslands"). In the backdune, where the substrate is more stable and richer in organic matter, a chamaephytic vegetation occurs, known as the *Centaureo sphaerocephalae-Ononidetum ramosissimae*. In Sicily, these psammophilous communities are quite widely represented [11,46] despite their bad conservation status.

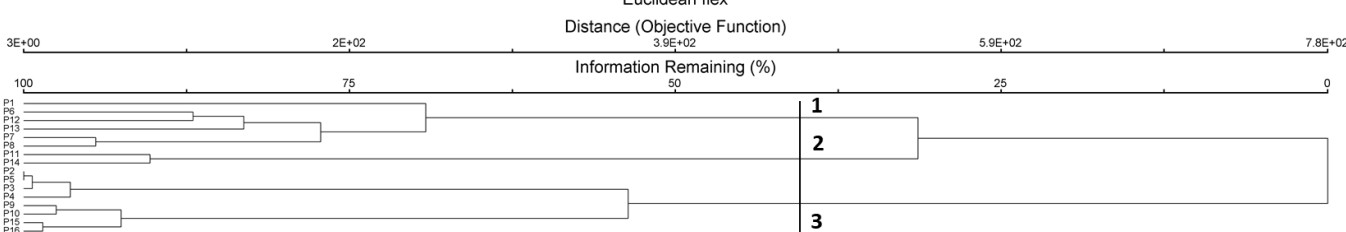

**Figure 2.** Cluster analysis of coastal dune communities. Plant communities: 1. *Cypero capitati-Agropyretum juncei*; 2. *Centaureo sphaerocephalae-Ononidetum ramosissimae*; 3. *Sileno coloratae-Ononidetum variegatae*.

### 3.3.2. Rocky Coast Vegetation (Figures 3 and 4; Table S2)

The coastal rocky outcrops of the Manghisi peninsula are colonized by halophytic communities belonging to the classes *Salicornietea fruticosae*, *Crithmo-Staticetea*, and *Saginetea maritimae*. The first belt, closest to the sea and subjected to salt spray, as well as the brackish rocky pools periodically inundated behind the cliffs, are home to a vegetation dominated by *Arthrocaulon meridionale*, a succulent chenopod shrub usually linked to seasonally inundated saltmarshes, and by *Limonium virgatum* and *Limbarda crithmoides*. This vegetation is attributable to the *Limonio virgati-Arthrocnemetum macrostachyi*, an association belonging to the class *Salicornietea fruticosae*, described by Biondi et al. [47] from southern Apulia and already reported for Sicily by Minissale and Sciandrello [11]. Immediately inwards follows the *Limonietum syracusani* (habitat 1240 "Vegetated sea cliffs of the Mediterranean coasts with endemic *Limonium* spp."), a perennial association of the *Crithmo-Staticetea* and dominated by *Limonium syracusanum* and *Crithmum maritimum*, with a few other halophilous/subalophilous species. On corroded surfaces, more or less flat, with thin soil, subjected to disturbance, a prostrate perennial vegetation dominated by *Lotus cytisoides* and *Frankenia hirsuta* develops, taking the place of the *Limonietum syracusani*. The subhalophilous ephemeral communities, characterized by small therophytic, xerophilous plants with a short spring cycle growing in the small limestone rocky pools covered by a thin layer of sandy–loamy soil rich in salts and nitrates and periodically inundated by salt water, are referred to *Parapholido incurvae-Spergularietum marinae*, a new association described here (Holotypus: rel.2, Table 1), characterized by the dominance of *Parapholis incurva* and

*Spergularia marina* (=*S. salina*) and belonging to the *Sileno sedoidis-Catapodion balearici* alliance (class *Saginetea maritimae*, order *Saginetalia maritimae*). This annual spring vegetation is in catenal contact with the perennial communities belonging to the *Crithmo-Staticetea* class.

**Table 1.** *Parapholido incurvae-Spergularietum marinae* ass. nova hoc loco—sampling plots of main features of plant community investigated. According to the Braun-Blanquet method in each relevé the complete list of vascular plant species was recorded and for each species the cover value (percentage of soil surface) was assessed (+: <1% cover; 1: 1–5% cover; 2: 5–25% cover; 3: 25–50% cover; 4: 50–75% cover; 5: >75% cover). Asterisk (*) indicates the holotypus of the new association.

| Relevé number | 1 | 2 * | 3 | 4 | 5 | 6 | 7 | 8 | 9 | 10 | 11 | |
|---|---|---|---|---|---|---|---|---|---|---|---|---|
| Original relevé number | 17 | 18 | 19 | 20 | 21 | 22 | 23 | 24 | 25 | 26 | 27 | |
| Number Cluster | 4 | 4 | 4 | 4 | 4 | 4 | 4 | 4 | 4 | 4 | 4 | |
| Surface (mq) | 2 | 2 | 3 | 3 | 2 | 2 | 2 | 2 | 2 | 2 | 2 | |
| Coverage (%) | 60 | 70 | 80 | 60 | 100 | 80 | 70 | 85 | 85 | 85 | 85 | |
| Altitude (m a.s.l.) | 2 | 3 | 2 | 3 | 2 | 2 | 8 | 6 | 6 | 6 | 6 | |
| Floristic richness | 6 | 8 | 10 | 4 | 11 | 14 | 4 | 7 | 6 | 7 | 7 | presence |
| **Char. Ass.** | | | | | | | | | | | | |
| *Parapholis incurva* (L.) C.E. Hubb. | 2 | 2 | 3 | 3 | 3 | 2 | 2 | 4 | 3 | 4 | 3 | 11 |
| **Char. Sileno sedoidis-Catapodion balearici de Foucault and Bioret 2010 corr. Tomaselli et al. 2020** | | | | | | | | | | | | |
| *Catapodium balearicum* (Willk.) H. Scholz | + | 1 | . | . | . | . | . | . | . | . | . | 2 |
| *Silene sedoides* Poir. | . | + | . | . | . | . | . | . | . | . | + | 2 |
| **Char. Saginetea maritimae Westhoff, Van Leeuwen and Adriani 1962, Saginetalia maritimae Westhoff, Van Leeuwen & Adriani 1962** | | | | | | | | | | | | |
| *Spergularia marina* (L.) Besser | 3 | 4 | + | 1 | + | 3 | 1 | 1 | + | 1 | 2 | 11 |
| *Beta vulgaris* subsp. *maritima* (L.) Arcang. | + | + | 1 | . | + | 1 | . | 1 | + | 1 | . | 8 |
| *Medicago littoralis* Rohde ex Loisel. | + | + | + | . | 3 | 1 | . | . | . | + | + | 7 |
| *Plantago coronopus* L. | + | 1 | 1 | 1 | . | . | . | . | . | . | . | 4 |
| *Mesembryanthemum nodiflorum* L. | . | . | 1 | . | . | . | . | + | . | . | . | 2 |
| *Matthiola tricuspidata* (L.) W.T. Aiton | . | . | . | . | . | + | . | . | . | . | . | 1 |
| **Other species** | | | | | | | | | | | | |
| *Lotus cytisoides* L. | . | . | + | 1 | + | . | + | + | 1 | + | . | 7 |
| *Frankenia hirsuta* L. | . | . | + | . | + | . | 2 | 1 | 1 | + | 1 | 7 |
| *Trifolium resupinatum* L. | . | 1 | . | . | + | + | . | + | + | + | + | 7 |
| *Anthemis arvensis* L. | . | . | 1 | . | 1 | + | . | . | . | . | . | 3 |
| *Silene colorata* Poir. | . | . | . | . | + | + | . | . | . | . | . | 2 |
| *Glaucium flavum* Crantz | . | . | . | . | + | + | . | . | . | . | . | 2 |
| *Crithmum maritimum* L. | . | . | . | . | . | + | . | . | . | . | . | 1 |
| *Plantago macrorhiza* Poir. | . | . | . | . | . | + | . | . | . | . | . | 1 |
| *Trifolium nigrescens* Viv. | . | . | . | . | . | + | . | . | . | . | . | 1 |
| *Medicago truncatula* Gaertn. | . | . | + | . | . | . | . | . | . | . | . | 1 |
| *Lotus ornithopodioides* L. | . | . | . | . | . | + | . | . | . | . | . | 1 |
| *Reichardia picroides* (L.) Roth | . | . | . | . | + | . | . | . | . | . | . | 1 |
| *Sonchus asper* (L.) Hill | . | . | . | . | . | + | . | . | . | . | . | 1 |
| *Tamarix africana* Poir. | . | . | . | . | . | . | . | . | . | . | + | 1 |

### 3.3.3. Coastal Wetland Vegetation (Figure 5; Table S3)

Helophytic and Herbaceous Perennial Communities of Fresh and Brackish Waters

The wetland areas close to the industrial area, subjected to long periods of submersion, are covered by perennial vegetation dominated by helophytes, known as the class *Phragmito-Magnocaricetea*. In particular, according to the flooding period and water depth, the following zonation has been observed: in deeper waters (60–80 cm), the *Typhetum domingensis*; in shallow waters (50–60 cm), the *Eleocharido-Alismetum lanceolati*, characterized by the dominance of *Eleocharis palustris* and *Alisma lanceolatum*; in correspondence of small ponds in proximity of the sea, with 40–50 cm water depth, drying in summer, with clayey-sandy soils, the *Bolboschoeno-Alismetum lanceolati* occurs, a new association described here (Holotypus: rel. 6, Table 2), characterized by the dominance of *Alisma lanceolatum* and *Bolboschoenus maritimus*, and which can be referred to the *Glycerio-Sparganion neglecti* alliance (*Nasturtio-Glycerietalia fluitantis* order). The edges of the "Saline di Priolo" are covered by monophytic vegetation dominated by *Phragmites australis*, belonging to the *Phragmitetum communis*. This latter association is widespread in Sicily along both the coastal strip and the middle and final stretch of watercourses, where there is stagnant water with a certain degree of eutrophication. Often, when favored by waters rich in nitrates, it replaces the natural halophilic vegetation of the *Salicornietea fruticosae* class [5].

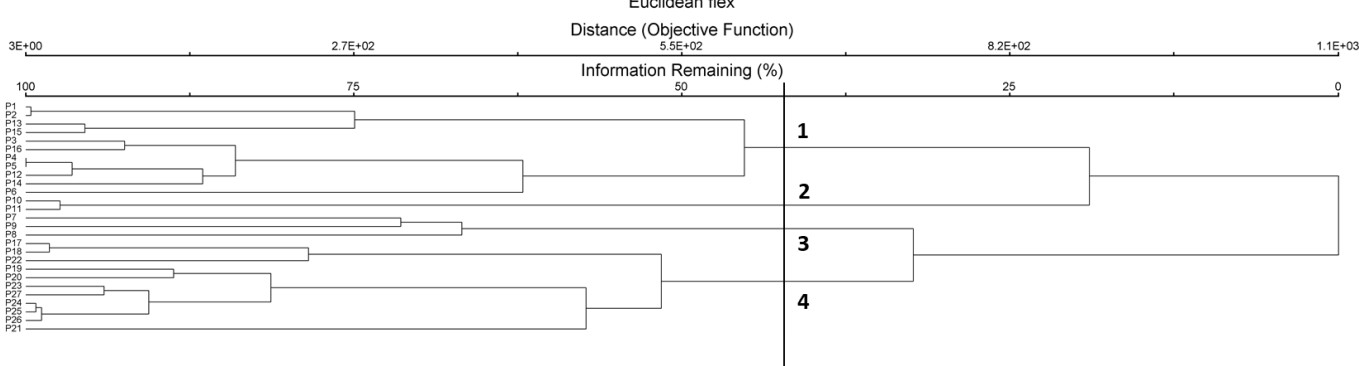

**Figure 3.** Cluster analysis of rocky coast communities. Plant communities: 1. *Limonietum syracusani*; 2. *Limonio-Arthrocnemetum macrostachyi*; 3. *Lotus cytisoides* and *Frankenia hirsuta* comm.; 4. *Parapholido incurvae-Spergularietum marinae* ass. nova.

The edges of depressed areas, with clayey–sandy soils periodically flooded, develop a perennial herbaceous vegetation enriched by floristic elements typical of the *Molinio-Arrhenatheretea* class, such as *Lythrum junceum*, *Potentilla reptans*, *Juncus articulatus*, *Oenanthe globosa*, *Trifolium fragiferum*, *Euphorbia hirsuta*, *Teucrium scordium*, *Kickxia commutata*, *Phyla nodiflora*, etc. Due to its ecological and floristic features, a new association with the name *Euphorbio hirsutae-Lotetum preslii* is described here (Holotypus: rel. 2, Table 3), characterized by the dominance of *Lotus corniculatus* subsp. *preslii* and *Euphorbia hirta*, which is referred to the *Paspalo-Agrostion semiverticillati* alliance (*Paspalo-Heleochloetalia*).

The large wetland area of the "Saline di Priolo", permanently flooded by more or less deep salt and brackish waters, is characterized by the hydrophytic submerged communities of *Enteromorpho intestinalidis-Ruppietum maritimae*; this vegetation is characterized by the dominance of *Ruppia maritima*, which occasionally grows with green algae as *Enteromorpha intestinalis*. In the study area, these communities are quite frequent, favored by waters rich in nitrates from the surrounding agricultural fields, and are usually in catenal contact with the bank vegetation, generally represented by annual halophilous communities of the *Thero-Suaedetea* class (habitat 1310 "Annual pioneer vegetation in Salicornia and other species of muddy and sandy areas") and by perennial halophilous communities of the *Salicornietea fruticosae* class (habitat 1420 "Mediterranean and thermo-Atlantic grasslands and fruit groves (*Sarcocornetea fruticosi*)"). These latter communities are characterized by

the presence of perennial Amaranthaceae species, and differ in species composition and cover in relation to the flooding period, thus characterizing a typical zonation from the innermost to the outer part of the brackish basin. Usually, in the inner part area of the saltmarsh, subjected to long periods of submersion, the *Junco subulati-Sarcocornietum alpini* and the *Arthrocaulo meridionalis-Juncetum subulati* are found; in the more peripheral part of the saltmarsh, rarely subjected to submersion, the *Agropyro scirpei-Inuletum crithmoidis* and the *Halimiono-Suaedetum verae* communities occur. These latter are halo-nitrophilic communities that, at present, have a very scattered distribution in the study area. The innermost parts of the saltmarshes, drying up in the summer–autumn period, are populated by succulent annual species of Amarantaceae with a summer cycle, such as *Salicornia perennans* (=*Salicornia patula*) and *Suaeda maritima*, ascribable to the *Suaedo-Salicornietum patulae* association. This vegetation is typical of salty soils rich in organic matter. Usually, it alternates with *Ruppia maritima* communities over the growing season, with *Ruppietum maritimae* hydrophytic communities growing during the flooding period, and the *Suaedo-Salicornietum patulae* in the summer/autumn season, on the dried-up substrata. On soils rich in sandy–silty components and subjected to short periods of submersion, the presence of *Juncus acutus* communities is significant. This pythocoenosis comprises hygro-halophilous hemicryptophytes and geophytes, such as *Scirpoides holoschoenus*, *Juncus subulatus*, and *Carex extensa*, which allow us to refer, despite the absence of *Juncus maritimus*, to the *Juncetum maritimo-acuti*. Small marginal areas rarely subject to flooding, on sandy soils, are covered by an herbaceous dense vegetation dominated by *Elytrygia atherica*, a species that forms dense and paucispecific populations. In correspondence with depressed areas characterized by a periodic supply of sandy–silty material, a sub-halophilous arboreal vegetation characterized by *Tamarix africana* and *T. gallica* develops. In some stretches, this woody vegetation is enriched with halophilous species typical of the class *Salicornietea fruticosae*. In particular, the presence of *Limbarda crithmoides* allows us to ascribe this phytocoenosis to the *Inulo crithmoidis-Tamaricetum africanae*. This association, included in the *Tamaricion africanae* (*Nerio-Tamaricetea*), has already been reported for the saltmarshes of southeastern Sicily [5]. Moreover, in the "Saline di Priolo", close to the industrial area, grows a patch of woody vegetation dominated by *Ulmus minor*, probably of anthropogenic origin, in contact with the *Juncus acutus* community.

Temporary Ponds

On the carbonatic substrata of the Magnisi peninsula, in correspondence with small carbonate rocky pools with silty soil periodically flooded, this amphibious ephemeral vegetation occurs during the winter–spring period. From a phytosociological point of view, this vegetation type falls within the *Isoëto-Nanojuncetea* class, referable to the habitat 3170* "Mediterranean temporary ponds". The pools with shallow waters and thin soil, subjected to early drying in spring, are covered by amphibious vegetation dominated by *Bulliarda vaillantii* and *Lythrum hyssopifolia* and referrable to the *Lythro hyssopifoliae-Crassuletum vaillantii* association. In the nearby area of "Capo Murro di Porco", the association *Pulicario grecae-Damasonietum bourgaei* has been described, which is very similar to the *Lythro hyssopifoliae-Crassuletum vaillantii* community detected in the Magnisi peninsula due to the presence of *Damasonium bourgaei*.

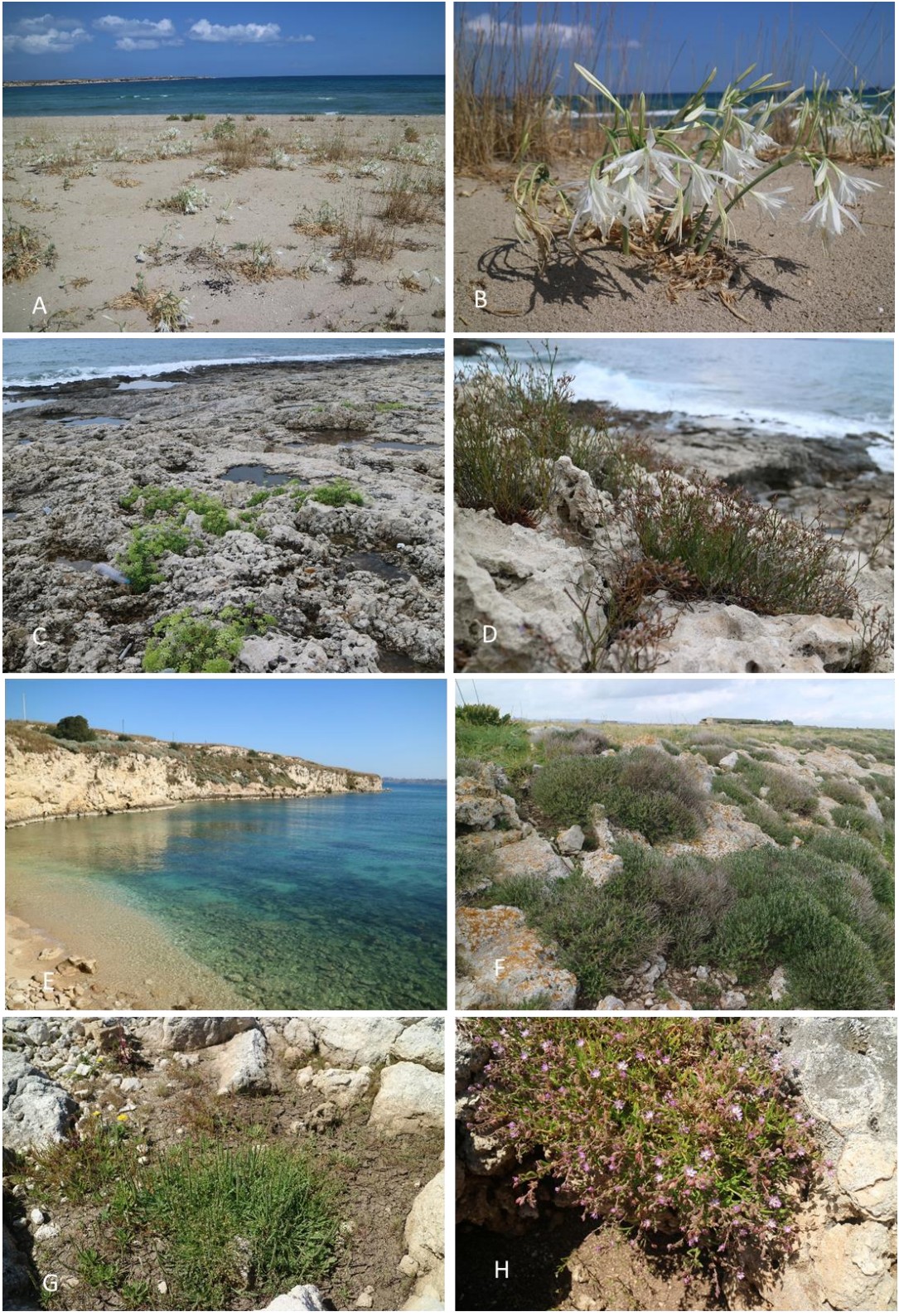

**Figure 4.** Photo plate illustrating different habitat types. (**A**,**B**) Psammophylous vegetation of the coast dunes (Saline di Priolo); (**C**,**D**) rocky coast vegetation with *Limonium syracusanum* and *Crithmum maritimum*. (Penisola di Magnisi); (**E**) rocky coast with *Artemisia arborescens* vegetation mixed with *Hyparrhenia hirta* dry grassaland; (**F**) Garrigues with *Tymbra capitata* (Penisola di Magnisi); (**G**,**H**) Subhalophilous ephemeral communities with *Spergularia marina* and *Parapholis incurva* (Penisola di Magnisi).

**Table 2.** *Bolboschoeno-Alismetum lanceolati* ass. nova hoc loco—sampling plots of main features of plant community investigated. According to the Braun-Blanquet method in each relevé the complete list of vascular plant species was recorded and for each species the cover value (percentage of soil surface) was assessed (+: <1% cover; 1: 1–5% cover; 2: 5–25% cover; 3: 25–50% cover; 4: 50–75% cover; 5: >75% cover). Asterisk (*) indicates the holotypus of the new association.

| Relevé number | 1 | 2 | 3 | 4 | 5 | 6 * | 7 | 8 | |
|---|---|---|---|---|---|---|---|---|---|
| Original relevé number | 22 | 23 | 24 | 26 | 27 | 30 | 31 | 32 | |
| Number Cluster | 4 | 4 | 4 | 4 | 4 | 4 | 4 | 4 | |
| Surface (mq) | 16 | 16 | 16 | 16 | 16 | 16 | 16 | 16 | |
| Coverage (%) | 80 | 75 | 85 | 90 | 100 | 90 | 90 | 90 | |
| Altitude (m a.s.l.) | 5 | 5 | 5 | 5 | 3 | 2 | 2 | 2 | |
| Floristic richness | 8 | 17 | 11 | 9 | 10 | 12 | 11 | 7 | presence |
| Char. Ass. | | | | | | | | | |
| *Bolboschoenus maritimus* (L.) Palla | 4 | 1 | 2 | 1 | 4 | 2 | 4 | 2 | 8 |
| Char. Glycerio-Sparganion neglecti Br.-Bl. and Sissing in Boer 1942 | | | | | | | | | |
| *Alisma lanceolatum* With. | 2 | 4 | 4 | 4 | 3 | 4 | 4 | 4 | 8 |
| Char. Phragmito-Magnocaricetea Klika in Klika and Novák 1941 | | | | | | | | | |
| *Phragmites australis* (Cav.) Trin. ex Steud. | + | . | . | 1 | . | + | 1 | 1 | 5 |
| *Veronica anagallis-aquatica* L. | + | 1 | + | . | . | + | . | + | 5 |
| *Eleocharis palustris* (L.) Roem. and Schult. | . | + | . | . | 1 | 1 | . | . | 3 |
| *Typha domingensis* (Pers.) Steud. | . | . | . | . | . | . | 1 | 1 | 2 |
| *Carex otrubae* Podp. | . | + | . | . | . | . | . | . | 1 |
| Transgr. Molinio-Arrhenatheretea R.Tx.1937 | | | | | | | | | |
| *Potentilla reptans* L. | . | + | + | + | + | 1 | + | . | 6 |
| *Lotus corniculatus* L. subsp. *preslii* (Ten.) P.Fourn. | . | 1 | + | + | . | 1 | 1 | . | 5 |
| *Lythrum junceum* Banks and Sol. | + | 1 | 1 | + | . | . | . | . | 4 |
| *Phyla nodiflora* (L.) Greene | . | . | . | . | . | 2 | 1 | 1 | 3 |
| *Juncus articulatus* L. | . | . | . | . | . | 2 | + | 1 | 3 |
| *Trifolium fragiferum* L. | . | + | + | . | . | . | . | . | 2 |
| Other species | | | | | | | | | 8 |
| *Symphyotrichum squamatum* (Spreng.) G.L. Nesom | . | + | + | + | + | + | + | . | 6 |
| *Mentha pulegium* L. | + | 1 | + | + | + | . | . | . | 5 |
| *Ranunculus trilobus* Desf. | . | + | 1 | + | + | . | . | . | 4 |
| *Tamarix africana* Poir. | 1 | . | . | . | 1 | . | 1 | . | 3 |
| *Plantago media* L. | . | + | . | . | . | + | + | . | 3 |
| *Holcus lanatus* L. | . | 1 | + | . | . | . | . | . | 2 |
| *Juncus hybridus* Brot. | . | + | . | . | . | . | . | . | 1 |
| *Isolepis cernua* (Vahl) Roem. and Schult. | . | . | . | . | . | + | . | . | 1 |
| *Juncus acutus* L. | . | . | . | . | + | . | . | . | 1 |
| *Carex extensa* Gooden. | . | + | . | . | . | . | . | . | 1 |
| *Vitex agnus-castus* L. | . | . | . | . | + | . | . | . | 1 |
| *Rumex pulcher* L. | . | + | . | . | . | . | . | . | 1 |
| *Chara* sp. | 1 | . | . | . | . | . | . | . | 1 |

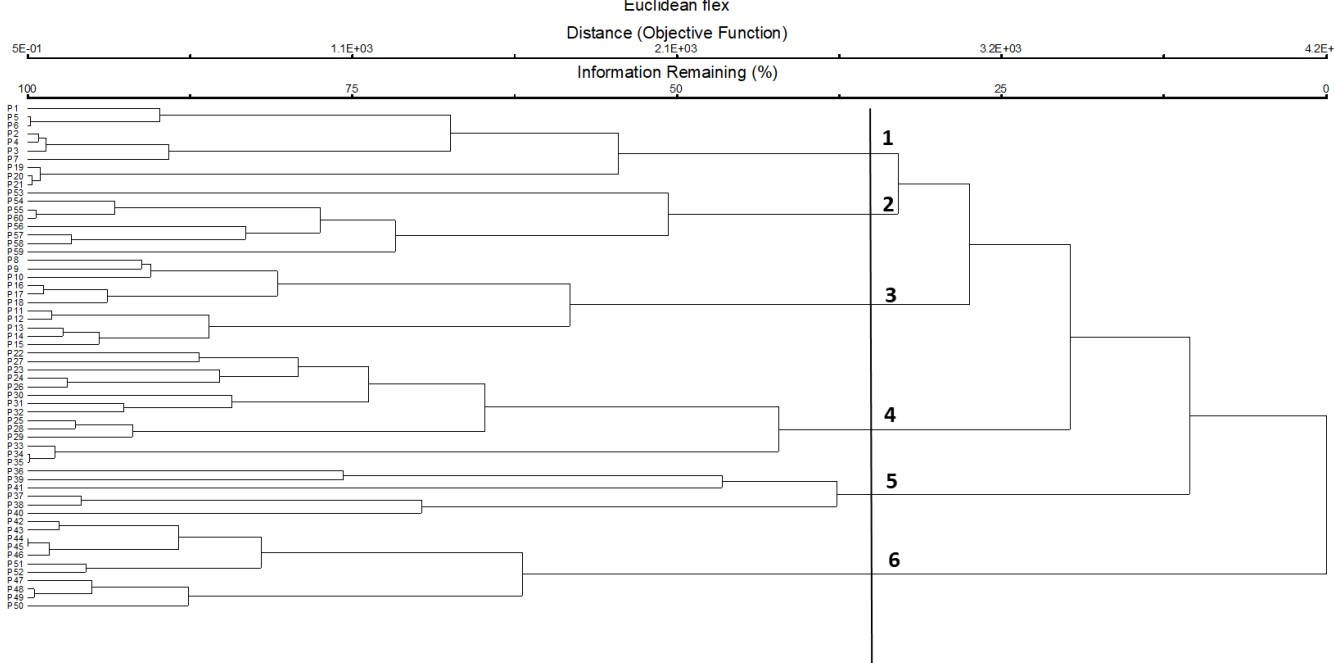

**Figure 5.** Cluster analysis of coastal wetland vegetation. Plant communities: 1. *Junco subulati-Sarcocornietum alpini/Arthrocaulo meridionalis-Juncetum subulati/Suaedo-Salicornietum patulae*; 2. *Lythro hyssopifoliae-Crassuletum vaillantii*; 3. *Agropyro scirpei-Inuletum crithmoidis/Halimiono-Suaedetum verae*; 4. *Phragmition/Scirpion compacti/Glycerio-sparganion*; 5. *Euphorbio hirsutae-Lotetum preslii/Juncus acutus* comm.; 6. *Limbardo crithmoidis-Tamaricetum africanae*.

**Table 3.** *Euphorbio hirsutae-Lotetum preslii* ass. nova hoc loco—sampling plots of main features of plant community investigated. According to the Braun-Blanquet method in each relevé the complete list of vascular plant species was recorded and for each species the cover value (percentage of soil surface) was assessed (+: <1% cover; 1: 1–5% cover; 2: 5–25% cover; 3: 25–50% cover; 4: 50–75% cover; 5: >75% cover). Asterisk (*) indicates the holotypus of the new association.

| Relevé number | 1 | 2 * | |
|---|---|---|---|
| Original relevé number | 40 | 41 | |
| Number Cluster | 5 | 5 | |
| Surface (mq) | 16 | 16 | |
| Coverage (%) | 100 | 90 | |
| Altitude (m a.s.l.) | 2 | 6 | |
| Floristic richness | 18 | 20 | presence |
| Char. Ass. | | | |
| *Lotus corniculatus* L. subsp. *preslii* (Ten.) P.Fourn. | 3 | 3 | 2 |
| *Euphorbia hirsuta* L. | 2 | 2 | 2 |
| Char. Paspalo-Agrostion semiverticillati Br.-Bl. in Br.-Bl. Roussine and Negre 1952 and Paspalo-Heleochloetalia Br.-Bl. ex Rivas Goday 1956 | | | |
| *Symphyotrichum squamatum* (Spreng.) G. L. Nesom | 1 | + | 2 |
| Char. Molinio-Arrhenatheretea R.Tx.1937 | | | |
| *Phyla nodiflora* (L.) Greene | 1 | 3 | 2 |
| *Lythrum junceum* Banks & Sol. | 3 | 1 | 2 |
| *Potentilla reptans* L. | + | 2 | 2 |
| *Juncus articulatus* L. | 1 | + | 2 |
| *Scirpoides holoschoenus* (L.) Soják | 1 | + | 2 |
| *Teucrium scordium* L. | + | + | 2 |
| *Oenanthe globulosa* L. | . | + | 1 |
| *Kickxia commutata* (Bernh. ex Rchb.) Fritsch | . | 1 | 1 |
| *Trifolium resupinatum* L. | + | . | 1 |

**Table 3.** *Cont.*

| Other species | | | |
|---|---|---|---|
| *Bolboschoenus maritimus* (L.) Palla | + | + | 2 |
| *Dipsacus fullonum* L. | + | . | 1 |
| *Rubus ulmifolius* Schott | 1 | . | 1 |
| *Tamarix africana* Poir. | 1 | . | 1 |
| *Schenkia spicata* (L.) G. Mans. | . | 2 | 1 |
| *Cynodon dactylon* (L.) Pers. | . | 2 | 1 |
| *Daucus carota* L. subsp. *maritimus* (Lam.) Batt. | + | . | 1 |
| *Xanthium italicum* Moretti | . | 1 | 1 |
| *Ranunculus trilobus* Desf. | 1 | . | 1 |
| *Mentha pulegium* L. | . | 1 | 1 |
| *Plantago media* L. | . | 1 | 1 |
| *Carex extensa* Gooden. | 1 | . | 1 |
| *Juncus acutus* L. | 2 | . | 1 |
| *Phragmites australis* (Cav.) Trin. ex Steud. | . | + | 1 |
| *Alisma lanceolatum* With. | . | + | 1 |
| *Typha domingensis* (Pers.) Steud. | . | + | 1 |

### 3.3.4. Dry Grasslands and Garrigues/Shrubs (Figures 6 and 7; Table S4)

The degradation of the Mediterranean maquis with *Pistacia lentiscus* allows the expansion of garrigues with *Tymbra capitata* or grassland dominated by *Hyparrhenia hirta* (habitat 6220, "Pseudo-steppe with grasses and annuals of the Thero-Brachypodietea"). From a phytosociological point of view, the grassland is very rich in thermo-xerophilous floristic elements typical of the *Hyparrhenion hirtae* alliance, such as *Hyparrenia hirta*, *Andropogon distachyos*, *Asphodelus ramosus*, *Ferula communis*, *Lathyrus clymenum*, *Thapsia garganica*, *Foeniculum vulgare* subsp. *piperitum*, *Carlina corymbosa*, *Daucus carota*, *Eryngium campestre*, *Moraea sisyrinchium*, *Lobularia maritima*, etc. The habitat includes arid Mediterranean grasslands, typical of shallow oligotrophic soils, characterized by a high number of hemicryptophyte species, generally mixed with annual herbaceous species. This environment is also occupied by therophytic spring vegetation, linked to small layers of soil placed on rocky ledges or in small corrosion pools, and usually forming a mosaic with the perennial herbaceous vegetation or garrigues. It is dominated by the presence of some annual Crassulaceae, such as *Sedum caeruleum*, *S. stellatum*, *Crassula tillaea*, etc. This plant community is referrable to the *Thero-Sedetum caerulei* association, a pioneer coenosis in contact with garrigue vegetation dominated by *Tymbra capitata*. On the Magnisi peninsula, the *Tymbra capitata* vegetation occupies small and limited areas, on carbonate rocky outcrops, due to the frequent summer fires which hinder the evolutionary process of these chamaephytic communities towards the evergreen Mediterranean shrub. For these reasons, there are few diagnostic species of *Cisto-Micormerietea*, such as *Micromeria graeca* subsp. *tenuifolia*, *Poterium spinosum*, and *Phagnalon saxatile*. The rocky slopes facing west of the Magnisi peninsula are home to communities dominated by *Artemisia arborescens* that, for their ecological, structural, and floristic features, can be referred to the *Atriplici halimi-Artemisietum arborescentis* association [48], despite the lack of the characteristic species *Atriplex halimus*. This vegetation is in contact towards the coast with the halophilous coastal vegetation of the *Crithmo-Limonieta*, while landwards, it is contact with the *Hyparrhenia* steppe grasslands. It can be considered as a permanent edaphic halo-nitrophilous community whose evolution is prevented by anthropic disturbance, combined with the action of coastal winds. Similar communities have been found on Lachea island, near Catania. Furthermore, the peninsula is affected by extensive annual meadows dominated by *Stipellula capensis*, mainly favored by summer fires and by perennial herbaceous formations dominated by *Oloptum miliaceum*, linked to grazing.

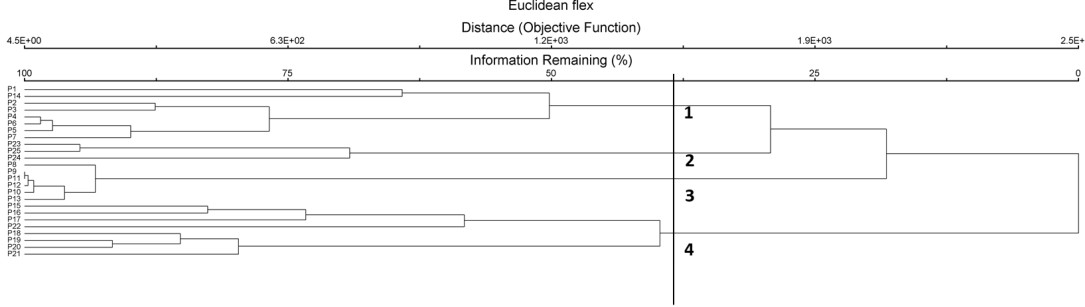

**Figure 6.** Cluster analysis of dry grasslands, garrigues, and shrubs. Plant communities: 1. *Artemisia arborescens* comm./*Hyparrhenietum hirto-pubescentis*/*Thymbra capitata* comm.; 2. *Thero-Sedetum caerulei*; 3. *Oloptum miliaceum* comm.; 4. *Stipellula capensis* comm.

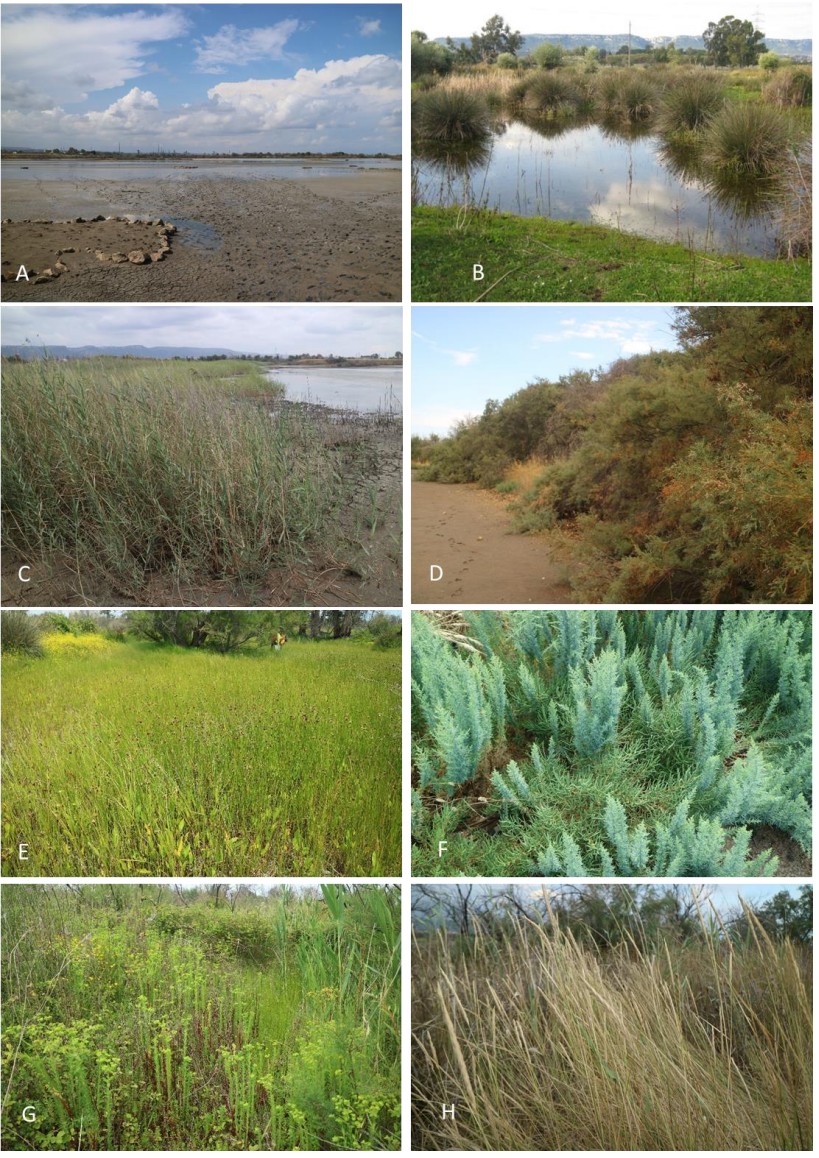

**Figure 7.** Photo plate illustrating different habitat types. (**A**) Saline di Priolo during the summer; (**B**) *Juncus acutus* community (Saline di Priolo); (**C**) *Phragmites australis* community in contact with wetlands of Saline di Priolo; (**D**) *Tamarix* sp pl. vegetation; (**E**) *Bolboschoeno-Alismetum lanceolati* ass. nova (Saline di Priolo); (**F**) *Suaedo-Salicornietum patulae* (Saline di Priolo); (**G**) *Euphorbio hirsutae-Lotetum preslii* ass. nova (Saline di Priolo); (**H**) *Elymetum atherici* (Saline di Priolo).

### 3.4. Diacronic Analysis—Landscape Composition and Configuration

The photointerpretation of the oldest aerial photos (1955) allowed the identification of 10 main types of plant communities, which is a limited number of classes if compared to the number of types (19) derived from the photointerpretation of the recent aerial photos (2015) (Figures 8 and 9). This discrepancy is partly due to a greater spatial and thematic resolution of the aerial images than of the map in 2015, but also to the presence of artificial, synanthropic, and semi-natural classes which were not present in 1955, the landscape being completely natural at that time. In fact, the comparison of the two periods highlights that the anthropogenic transformations and the almost complete occupation of the coastal environments by industrial settlements have caused a sharp reduction in the dune and wetland systems, due to the leveling and removal of sand and to the reclamation activities (Table 4). In particular, the results show a reduction in the aquatic vegetation (*Ruppion maritimae*) of the Saline di Priolo, decreasing from 85 ha (22.6%) in 1955 to only 26 ha (7.4%) in 2015, and the disappearance of the Priolo salt-works, an ancient system for the production of sea salt from sea water, which once occupied an area of 5 hectares. As is well-known, salt-works and salt pans are anthropic modifications of coastal lagoons and, although of anthropic origin, they maintain some of the floristic and faunal peculiarities of the natural systems. The saltmarsh communities (*Juncetea maritimi*, *Salicornietea fruticosae*), in catenal contact with aquatic vegetation, have also undergone a strong reduction from 93 ha (37%) in 1955 to only 25 ha (8%) in 2015. The annual habitats of the coastline (*Euphorbion peplidis/Maresion nanae*) and perennial dune habitat characterized by *Elytrigia juncea* (*Elytrigienion junceae*) suffered a reduction from 40 ha (11%) to 12 ha (3.5%). The industrial transformation of the area has been a very important driver of changes affecting the coastal environments, mainly because of the massive installation, covering a surface of 96 ha (27.3%); it is the widest land-use typology of the studied area together with reforestation, at 44 ha (12.6%), and uncultivated/abandoned lands colonized by pioneer grassland vegetation (*Echio-Galactition*), with an area of 96 ha (27%). A surprising piece of data concerns the current extension of the woodland cover (about 7 ha), in the Saline di Priolo, of *Tamarix* sp. pl. vegetation, which in the past was not present, probably due to the high salt concentrations in the soil linked to the activities of salt extraction (salt-works).

Table 5 shows the results of the landscape metrics applied to the two habitat maps. Two habitat types identified in 1955 are not shown in the 2015 map, either because they disappeared (2110) or because they were so reduced that they could not be graphically represented (habitat cover less than minimum mappable area). On the other hand, habitat types 1430, 3140, 5220, and 6220 reported in the 2015 habitat map are not present in the 1955 map due to the mere fact that they often cover limited surfaces and do not have a basic cartographic product with sufficient resolution, so it was not possible to identify them. Habitat 9320, detected in 2015, was not present in 1955. For all these reasons, we decided to concentrate our considerations on the habitats strictly relevant to the coastal strip and salt-marshes (except 1310), namely 1150, 1210, 1240, 1410, and 1420.

The values for CA and NumP are shown (Figure 10A,B). It is quite evident, as already mentioned above, that there has been a strong decrease in surface area for all habitat types under consideration (particularly dramatic for habitat 1420), together with a high fragmentation (increase in the number of patches, particularly evident in 1240). This result is in accordance with the values of MPS (Figure 10C), which shows a strong decrease for all habitat types (but particularly evident in 1210, 1240, and 1420), a trend resulting from the sharp reduction and fragmentation process of coastal environments. As is well-known, the Shape index describes the ratio between the perimeter of the patch and the square root of the patch area; AWMSI equals the average shape index (SHAPE) of patches of the corresponding patch type, weighted by patch area. In general, values increase as the shapes of patches become more complex [49]. In Figure 10D,E, AWMSI values are shown; no major differences are detectable between the two years of observation, which means that, despite the ongoing dynamics, the spatial complexity in individual classes is more or less the same. MPAR is another measure of shape complexity, but because it is not standardized to a

certain shape (e.g., a square), it describes the patch complexity in a straightforward way. Also, in this case, no particular changes can be evidenced between 1955 and 2015, except for habitat 1240, with a sharp increase in MPAR, probably linked to the fragmentation of the habitat into numerous small patches of irregularly linear shape. The case study of the Saline di Priolo is a paradigmatic example of how, in just a few decades, it is possible to destroy entire stretches of coastal environments with negative consequences on the structure and composition of coastal habitats and vascular flora. The comparison of aerial photos over a period of 60 years shows a dramatic change in the natural landscape, with a permanent loss of land in correspondence with the industrial settlements, the almost complete leveling of the dune system, and the extensive wetland reclamation (Figure 11). Another similar dramatic example of extensive destruction of coastal environments due to industrial installations has occurred in Macconi di Gela, along the southern coast of Sicily, with sharp changes in the dune complexes and wetlands [46].

### 3.5. Conservation Status

The floristic vegetation survey activities and cartographic processing have highlighted fifteen habitat types according to the Habitat Directive (Table 6), of which four are of priority conservation interest (1150, 3170, 5220, 6220). Of these fifteen, five habitat types turned out to be in the favorable conservation status, nine in the inadequate conservation status, and three in the bad conservation status, the latter one being a priority conservation habitat (5220* Zyziphus arborescent matorrals) with a decreasing trend. Furthermore, habitat 2210, "Fixed dunes of the coast (*Crucianellion maritimae*)", reported in the Natura 2000 sheet of the "Saline di Priolo" SAC, is considered to have disappeared or have been previously erroneously reported. Furthermore, the habitats 1410, "Mediterranean salt meadows (*Juncetalia maritimi*)", and 1420, "Mediterranean and thermo-Atlantic halophilous scrubs (*Sarcocornietea fruticosi*)", which once covered larger areas, are now very reduced and fragmented, resulting in an inadequate conservation status. Our results regarding the dune system, in line with Prisco et al. [50], highlighted the dramatically bad conservation status of dune habitats.

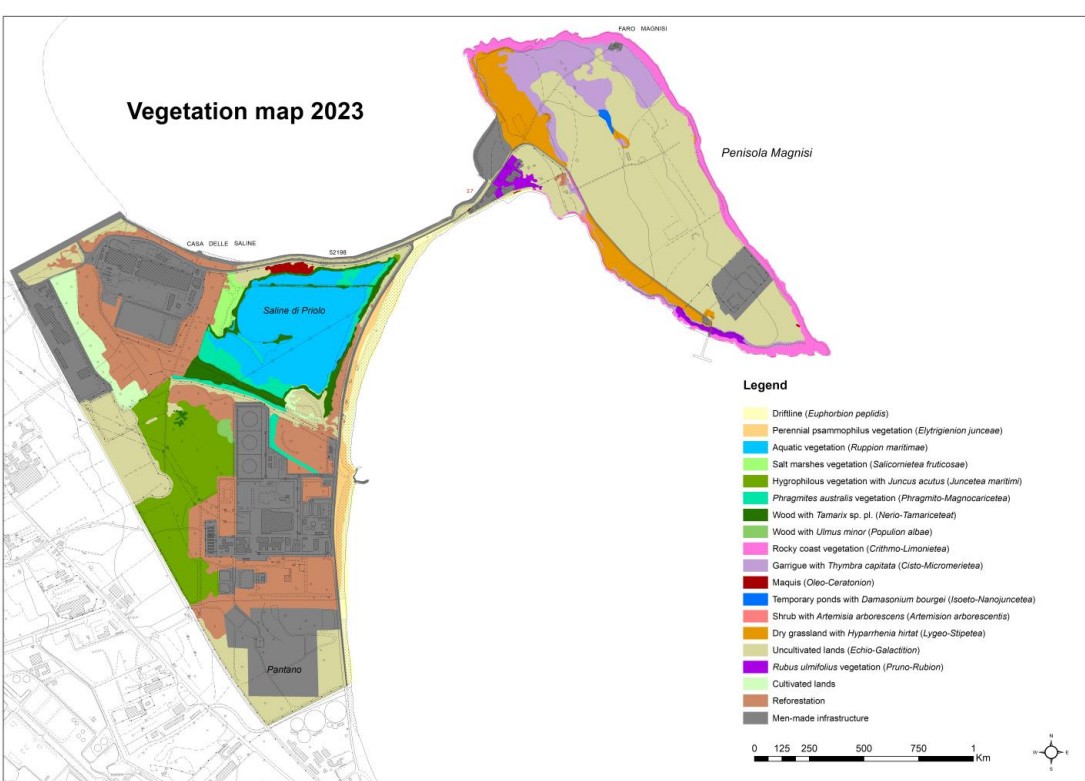

**Figure 8.** Current vegetation map of "Saline di Priolo and Penisola di Magnisi" (2023).

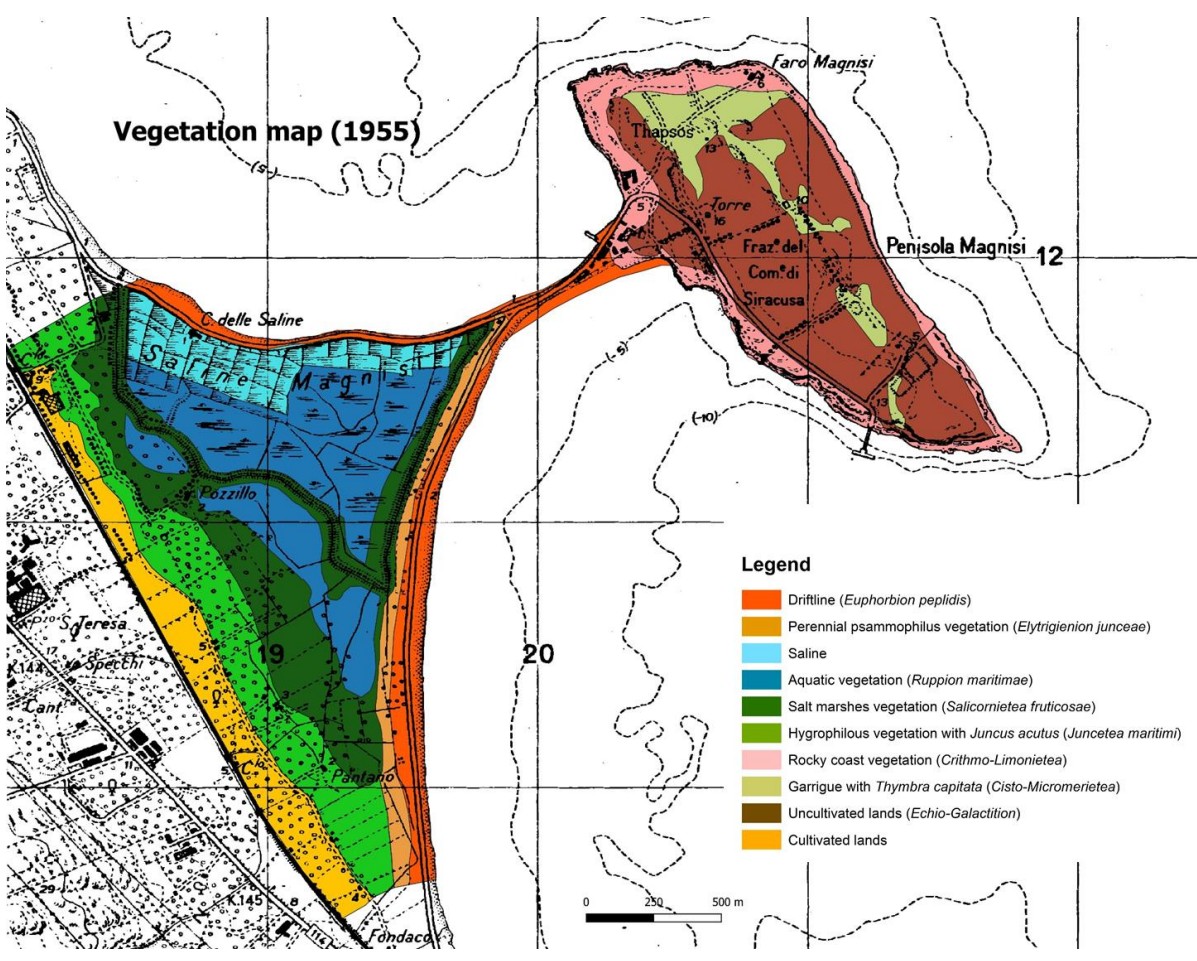

**Figure 9.** Historical vegetation map of "Saline di Priolo and Penisola di Magnisi" (1955).

**Table 4.** Surface comparison from the historical stereophoto (1955) and current aerial photos (2023).

|  | 1955 | % | 2023 | % |
|---|---|---|---|---|
| Driftline (*Euphorbion peplidis*) | 29 | 7.8 | 9 | 2.6 |
| Shifting dunes (*Elytrigienion junceae*) | 11 | 2.8 | 4 | 1.2 |
| Saline | 19 | 5.2 | 0 | 0.0 |
| Aquatic vegetation (*Ruppion maritimae*) | 85 | 22.6 | 25.9 | 7.4 |
| Salt-marsh vegetation (*Salicornietea fruticosae*) | 53 | 14.1 | 2.2 | 0.6 |
| Hygrophylous vegetation with *Juncus acutus* (*Juncetea maritimi*) | 40 | 10.6 | 23 | 6.5 |
| *Phragmites australis* vegetation (*Phragmito-Magnocaricetea*) | 0 | 0.0 | 5.89 | 1.7 |
| Woods with *Tamarix* sp. pl. (*Nerio-Tamaricetea*) | 0 | 0.0 | 6.6 | 1.9 |
| Woods with *Ulmus minor* (*Populion albae*) | 0 | 0.0 | 0.557 | 0.2 |
| Rocky coast vegetation (*Crithmo-Limonietea*) | 28 | 7.4 | 11.35 | 3.2 |
| Garrigue with *Thymbra capitata* (*Cisto-Micromerietea*) | 16.13 | 4.3 | 14.31 | 4.1 |
| Maquis with *Ziziphus lotus* (*Oleo-Ceratonion*) | 0 | 0.0 | 0.87 | 0.2 |
| Shrub with *Artemisia arborescens* (*Artemision arborescentis*) | 0 | 0.0 | 0.117 | 0.0 |
| Temporary ponds with *Damasonium bourgaei* (*Isoeto-Nanojuncetea*) | 0 | 0.0 | 0.315 | 0.1 |
| *Hyparrhenia hirta* dry grassland (*Lygeo-Stipetea*) | 0 | 0.0 | 12.6 | 3.6 |
| Uncultivated lands (*Echio-Galactition*) | 67 | 17.8 | 96 | 27.3 |
| *Rubus ulmifolius* vegetation (*Pruno-Rubion*) | 0 | 0.0 | 3 | 0.8 |

**Table 4.** *Cont.*

|  | **1955** | **%** | **2023** | **%** |
|---|---|---|---|---|
| Cultivated lands | 28.15 | 7.5 | 6 | 1.7 |
| Reforestation | 0 | 0.0 | 44.4 | 12.6 |
| Man-made infrastructure | 0 | 0.0 | 96 | 27.3 |
|  | 376,274 |  | 362,261 |  |

**Table 5.** Summary table of the landscape metrics applied to the different habitats in the 1955 and 2015 maps; the grey rows refer to those habitat types which it was possible to detect cartographically only on one of the two dates and which therefore were not the subject of temporal analysis.

| Habitat Type | AWMSI | | MSI | | MPAR | | MPS | | NumP | | CA | | TLA |
|---|---|---|---|---|---|---|---|---|---|---|---|---|---|
|  | 1955 | 2015 | 1955 | 2015 | 1955 | 2015 | 1955 | 2015 | 1955 | 2015 | 1955 | 2015 | 2015 |
| 1150 | 1.74 | 2.23 | 1.62 | 1.88 | 1854.842 | 11289.33 | 17.3 | 8.51 | 3 | 3 | 54.9 | 25.54 | 352.55 |
| 1210 | 4.74 | 3.68 | 4.74 | 3.92 | 3083.973 | 82176.50 | 29.21 | 1.32 | 1 | 2 | 29.21 | 2.63 | 352.55 |
| 1240 | 5.31 | 6.23 | 5.31 | 2.82 | 3540.334 | 61161.51 | 27.8 | 1.26 | 1 | 9 | 27.8 | 11.34 | 352.55 |
| 1310 | 2.10 | / | 2.10 | / | 1676.777 | / | 19.45 | / | 1 | / | 19.45 | / | 352.55 |
| 1410 | 2.69 | 2.36 | 2.69 | 2.36 | 1495.051 | 1174.30 | 40.16 | 23.04 | 1 | 1 | 40.16 | 23.04 | 352.55 |
| 1420 | 3.94 | 3.94 | 2.55 | 3.37 | 2231.837 | 23406.60 | 26.06 | 1.09 | 2 | 2 | 52.12 | 2.19 | 352.55 |
| 1430 | / | 2.18 | | 2.18 | | 2264.90 | | 0.12 | | 1 | | 0.12 | 352.55 |
| 2110 | 3.67 | / | 3.67 | / | 3990.6974 | / | 10.5 | / | 1 | / | 11.7 | / | 352.55 |
| 3170 | / | 1.50 | | 1.50 | | 947.90 | | 0.31 | | 1 | | 0.31 | 352.55 |
| 5220 | / | 1.20 | | 1.20 | | 3427.80 | | 0.02 | | 1 | | 0.02 | 352.55 |
| 5420 | 2.70 | 3.33 | 1.84 | 3.19 | 3998.618 | 53336.03 | 5.4 | 1.79 | 3 | 8 | 16.13 | 14.30 | 352.55 |
| 6220 | / | 2.57 | | 1.94 | | 35172.79 | | 1.62 | | 11 | | 17.87 | 352.55 |
| 9320 | / | 1.96 | | 1.96 | | 5281.10 | | 0.02 | | 1 | | 0.02 | 352.55 |

**Table 6.** Conservation status and trend of the habitats of the "Saline di Priolo" SAC. FV—favorable; U1—inadequate; U2—bad; (=) trend stable; (-) trend decreasing; (x) trend unknown. Asterisk (*) indicates a priority habitat.

| Habitat | ha | % | Conservation Status and Trend |
|---|---|---|---|
| 1150 * Coastal lagoons | 25.9 | 17.7 | FV |
| 1210 Annual vegetation of drift lines | 9 | 6.1 | FV |
| 1240 Vegetated sea cliffs of the Mediterranean coasts with endemic Limonium spp. | 11.35 | 7.7 | U1 (-) |
| 1310 Salicornia and other annuals colonizing mud and sand | 0.1 | 0.1 | U1 (=) |
| 1410 Mediterranean salt meadows (Juncetalia maritimi) | 23 | 15.7 0.0 | U1 (x) |
| 1420 Mediterranean and thermo-Atlantic halophilous scrubs (Sarcocornetiea fruticosi) | 2.2 | 1.5 | U1 (x) |
| 1430 Halo-nitrophilous scrubs (Pegano-Salsoletea) | 0.12 | 0.1 | FV |
| 2110 Embryonic shifting dunes | 4 | 2.7 | U2 (=) |
| 2210 Crucianellion maritimae fixed beach dunes | ? | 0 | U2 (-) |
| 2230 Malcolmietalia dune grasslands | 0.2 | 0.1 | U1 (x) |
| 3170 * Mediterranean temporary ponds | 0.32 | 0.2 | U1 (=) |
| 5220 * Arborescent matorral with Zyziphus | 0.1 | 0.1 | U2 (-) |
| 5420 Sarcopoterium spinosum phryganas | 14.3 | 9.8 | FV |
| 6220 * Pseudo-steppe with grasses and annuals of the Thero-Brachypodietea | 12.6 | 8.6 | FV |
| 92D0 Southern riparian galleries and thickets (Nerio-Tamaricetea and Securinegion tinctoriae) | 6.6 | 4.5 | U1 (-) |
| 9320 Olea and Ceratonia forests | 0.8 | 0.5 | U1 (-) |
|  | 110.6 |  |  |

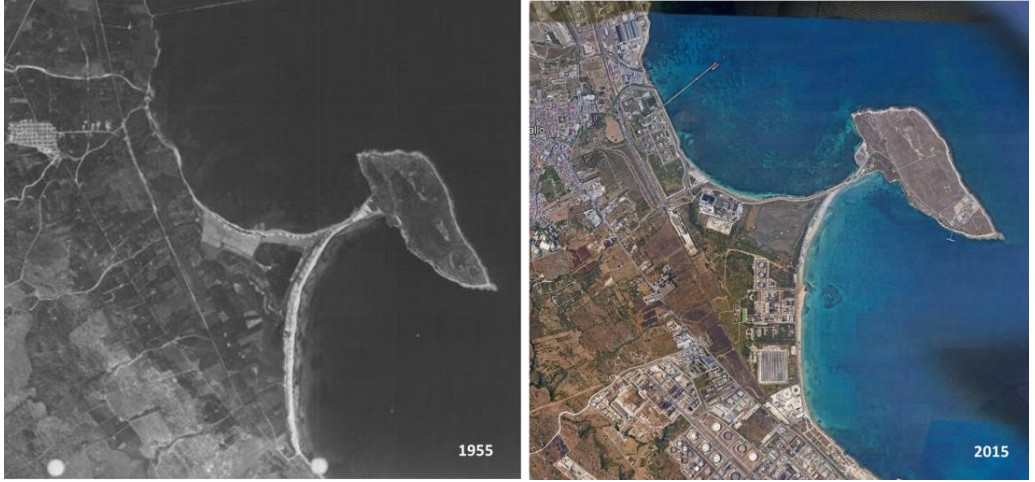

**Figure 10.** CA (Class Area) and NumP (Number of Patches) (**A**,**B**); MPS (Mean Patch Size) (**C**); AWMSI (Area-Weighted Mean Shape Size) and MPAR (Mean Perimeter–Area Ratio) (**D**,**E**) of habitat types 1150, 1210, 1240, 1410, and 1420 in 1955 and 2015.

**Figure 11.** Aerial photos of 1955 and 2015.

## 4. Conclusions

Vegetation analysis with the phytosociological method is basic for detecting and assessing the terrestrial habitats in line with the objectives of the European Habitats Directive 92/43/EEC. Although the study area is strongly altered by human pressures, the study allowed us to identify notable habitat diversity and floristic richness. As demonstrated by the diachronic analysis carried out, the area has undergone notable transformations, which occurred especially in the 1950s and 1960s with the industrialization of the area. Despite that, the wetland system is still able to support specialized flora, vegetation, and fauna, and provide meaningful ecosystem services. In conclusion, the knowledge of current and past vegetation acquired in this study can represent a valid basic tool for the protected area to plan targeted conservation actions for the most threatened and endangered habitats.

**Supplementary Materials:** The following supporting information can be downloaded at: https://www.mdpi.com/article/10.3390/land13010062/s1. Table S1: Coastal dune vegetation; Table S2: Rocky coast vegetation; Table S3: Coastal wetland vegetation; Table S4: Dry grasslands and garrigues/shrubs.

**Author Contributions:** Conceptualization, S.S.; methodology, S.S. and V.T.; investigation, S.S.; data curation, S.S.; data elaboration, S.S., V.R. and V.T.; writing—original draft preparation, S.S.; writing—review and editing, S.S., V.R. and V.T. All authors have read and agreed to the published version of the manuscript.

**Funding:** This research was financially supported by the research program (Line 3 Starting Grant Progetto HAB-VEG cod. 22722132172) funded by the University of Catania.

**Data Availability Statement:** Data is contained within the article.

**Acknowledgments:** The authors would like to acknowledge the reviewers for their valuable suggestions and comments. The authors thank Fabio Cilea, director of the Saline di Priolo protected area, for technical support, useful advice, and for making the photographic material available.

**Conflicts of Interest:** The authors declare no conflict of interest.

## Appendix A. Species List of the Vascular Plants Recorded from the "Saline di Priolo" SAC (SE Sicily)

**Table A1.** Chorology: Steno-Medit./Medit.—Mediterranean; Endem. sic.—Endemic to Sicily; Endem. Ital.—Endemic to Italy; Euro-Med.—Euro-Mediterranean; Medit.-Turan.—Mediterranean-Turanian; Medit.-Atl.—Mediterranean-Atlantic; Neotrop.—Neotropical; Paleotemp.—Paleotemperate; Cosmop.—Cosmopolitan; Eurasiat.—Eurasiantic; Circumbor.—Curcumboreal; Europ.-Caucas.—Europe-Caucasic; Paleotrop.—Paleotropical. Life form: T—Therophytes; Ch—Chamaephytes; H—Hemicryptophytes; G—Geophytes; P—Phanerophytes; NP—Nanophanerophytes; I—Hydrophytes.

| N. | Family | Corology | Life Form | Species | Saline Priolo | Penisola Magnisi |
|---|---|---|---|---|---|---|
| 1 | Asteraceae | Medit.-Atl. | Ch | *Achillea maritima* (L.) Ehrend. and Y.P. Guo | 1 | |
| 2 | Amaranthaceae | SW-Medit. | Ch | *Achyranthes sicula* (L.) All. | 1 | 1 |
| 3 | Rosaceae | Eurasiat. | H | *Agrimonia eupatoria* L. | 1 | |
| 4 | Lamiaceae | Euri-Medit. | T | *Ajuga chamaepitys* (L.) Schreb. | 1 | 1 |
| 5 | Alismataceae | Subcosmop. | I | *Alisma lanceolatum* With. | 1 | |
| 6 | Amaryllidaceae | Europ. | G | *Allium sphaerocephalon* subsp. *arvense* (Guss.) Arcang. | | 1 |
| 7 | Amaryllidaceae | Steno-Medit. | G | *Allium roseum* L. | 1 | |
| 8 | Amaryllidaceae | Steno-Medit. | G | *Allium commutatum* Guss. | | 1 |
| 9 | Amaryllidaceae | Steno-Medit. | G | *Allium subhirsutum* L. | 1 | |
| 10 | Asteraceae | Steno-Medit. | T | *Anacyclus clavatus* (Desf.) Pers. | | 1 |
| 11 | Poaceae | Paleotrop. | H | *Andropogon distachyos* L. | | 1 |
| 12 | Apiaceae | Euri-Medit. | H | *Anethum piperitum* Ucria | 1 | 1 |

**Table A1.** *Cont.*

| N. | Family | Corology | Life Form | Species | Saline Priolo | Penisola Magnisi |
|----|--------|----------|-----------|---------|---------------|------------------|
| 13 | Poaceae | Paleosubtrop. | T | *Anisantha rigida* (Roth) Hyl. | 1 | |
| 14 | Poaceae | Medit.-Turan. | T | *Anisantha rubens* (L.) Nevski | 1 | |
| 15 | Poaceae | Medit.-Turan. | T | *Anisantha sterilis* (L.) Nevski | 1 | |
| 16 | Poaceae | Paleotemp. | T | *Anisantha tectorum* (L.) Nevski | 1 | 1 |
| 17 | Asteraceae | Steno-Medit. | T | *Anthemis arvensis* L. | 1 | 1 |
| 18 | Apiaceae | Subcosmop. | T | *Torilis arvensis* (Huds.) Link | 1 | |
| 19 | Plantaginaceae | Endem. Ital. | Ch | *Antirrhinum siculum* Mill. | | 1 |
| 20 | Asteraceae | S-Medit. | NP | *Artemisia arborescens* (Vaill.) L. | | 1 |
| 21 | Asteraceae | Steno-Medit. | H | *Carlina corymbosa* L. | | |
| 22 | Lamiaceae | Steno-Medit. | T | *Stachys major* (L.) Bartolucci and Peru | | |
| 23 | Amaranthaceae | Medit. | Ch | *Arthrocaulon meridionale* Es.Ramírez, Rufo, Sánchez Mata, V.Fuente | 1 | 1 |
| 24 | Araceae | Steno-Medit. | G | *Arum italicum* Mill. | 1 | |
| 25 | Poaceae | Subcosmop. | G | *Arundo donax* L. | 1 | |
| 26 | Asparagaceae | Steno-Medit. | G | *Asparagus acutifolius* L. | 1 | 1 |
| 27 | Asphodelaceae | Subtrop. | H | *Asphodelus fistulosus* L. | 1 | 1 |
| 28 | Asphodelaceae | Steno-Medit. | G | *Asphodelus ramosus* L. | 1 | 1 |
| 29 | Asteraceae | Steno-Medit. | T | *Asteriscus aquaticus* (L.) Less. | | 1 |
| 30 | Fabaceae | S-Medit. | T | *Astragalus epiglottis* L. | | 1 |
| 31 | Fabaceae | S-Medit. | T | *Astragalus boeticus* L. | 1 | 1 |
| 32 | Fabaceae | Medit.-Turan. | T | *Astragalus hamosus* L. | | 1 |
| 33 | Fabaceae | Steno-Medit. | T | *Astragalus pelecinus* (L.) Barneby | | 1 |
| 34 | Amaranthaceae | Circumbor. | T | *Atriplex prostrata* Boucher ex DC. | 1 | |
| 35 | Poaceae | Medit.-Turan. | T | *Avena barbata* Pott ex Link | 1 | 1 |
| 36 | Poaceae | Medit.-Turan. | T | *Avena sterilis* L. | | 1 |
| 37 | Orobanchaceae | Euri-Medit. | T | *Bellardia trixago* (L.) All. | 1 | |
| 38 | Asparagaceae | Endem. Ital. | G | *Bellevalia dubia* (Guss.) Rchb. | | 1 |
| 39 | Asteraceae | Steno-Medit. | T | *Bellis annua* L. | | 1 |
| 40 | Amaranthaceae | Euri-Medit. | H | *Beta vulgaris* subsp. *maritima* (L.) Arcang. | 1 | 1 |
| 41 | Brassicaceae | Medit.-Turan. | T | *Biscutella didyma* L. | 1 | |
| 42 | Cyperaceae | Cosmop. | G | *Bolboschoenus maritimus* (L.) Palla | 1 | |
| 43 | Poaceae | Medit.-Turan. | T | *Brachypodium distachyon* (L.) P. Beauv. | 1 | 1 |
| 44 | Poaceae | Europ.-Caucas. | T | *Bromus racemosus* L. | | 1 |
| 45 | Crassulaceae | Subatl. | T | *Bulliarda vaillantii* (Willd.) DC. | | 1 |
| 46 | Apiaceae | Euri-Medit. | T | *Bupleurum tenuissimum* L. | 1 | |
| 47 | Brassicaceae | Medit.-Atl. | T | *Cakile maritima* Scop. | 1 | |
| 48 | Capparaceae | Steno-Medit. | NP | *Capparis orientalis* Veill. | 1 | 1 |
| 49 | Brassicaceae | Cosmop. | T | *Cardamine hirsuta* L. | 1 | |
| 50 | Asteraceae | Steno-Medit. | T | *Carduus argyroa* Biv. | | 1 |
| 51 | Asteraceae | Medit.-Turan. | H | *Carduus pycnocephalus* L. | 1 | |
| 52 | Cyperaceae | Steno-Medit. | G | *Carex hispida* Willd. ex Schkuhr | 1 | |
| 53 | Cyperaceae | Medit.-Atl. | H | *Carex extensa* Gooden. | 1 | |
| 54 | Cyperaceae | Atl. | H | *Carex otrubae* Podp. | 1 | |
| 55 | Cyperaceae | Eurosiber. | H | *Carex vulpina* L. | | 1 |
| 56 | Cyperaceae | Euri-Medit. | H | Carex distans L. | 1 | |
| 57 | Asteraceae | Endem. Ital. | H | *Carlina hispanica* Lam. | 1 | 1 |
| 58 | Asteraceae | Euri-Medit. | T | *Carthamus lanatus* L. | 1 | 1 |
| 59 | Poaceae | Medit.-Atl. | T | *Catapodium balearicum* (Willk.) H. Scholz | | 1 |
| 60 | Asteraceae | SW-Medit. | H | *Centaurea sicula* L. | | 1 |
| 61 | Asteraceae | Steno-Medit. | H | *Centaurea sphaerocephala* L. | 1 | |
| 62 | Gentianaceae | Paleotemp. | T | *Centaurium tenuiflorum* (Hoffmanns. and Link) Fritsch | 1 | |
| 63 | Characeae | | I | Chara sp. | 1 | |
| 64 | Asparagaceae | Steno-Medit. | G | *Squilla maritima* (L.) Steinh. | 1 | 1 |

**Table A1.** *Cont.*

| N. | Family | Corology | Life form | Species | Saline Priolo | Penisola Magnisi |
|---|---|---|---|---|---|---|
| 65 | Asteraceae | Cosmop. | H | *Cichorium intybus* L. | | 1 |
| 66 | Asteraceae | Orof. NE-Medit. | H | *Cirsium creticum* subsp. *triumfettii* (Lacaita) K. Werner | 1 | |
| 67 | Boraginaceae | Euri-Medit. | H | *Cynoglossum creticum* Mill. | 1 | 1 |
| 68 | Lamiaceae | Medit.-Mont. | Ch | *Clinopodium nepeta* (L.) Kuntze | | 1 |
| 69 | Convolvulaceae | Steno-Medit. | H | *Convolvulus althaeoides* L. | 1 | 1 |
| 70 | Convolvulaceae | Cosmop. | G | *Convolvulus soldanella* L. | 1 | |
| 71 | Asteraceae | Medit. | H | *Crepis bursifolia* L. | 1 | |
| 72 | Convolvulaceae | Cosmop. | Ch | *Cressa cretica* L. | 1 | |
| 73 | Apiaceae | Euri-Medit. | Ch | *Crithmum maritimum* L. | 1 | 1 |
| 74 | Convolvulaceae | Eurasiat. | T | *Cuscuta epithymum* (L.) L. | | 1 |
| 75 | Poaceae | Steno-Medit. | T | *Cutandia maritima* (L.) Benth. ex Barbey | 1 | |
| 76 | Poaceae | Steno-Medit. | T | *Cutandia divaricata* (Desf.) Barbey | 1 | |
| 77 | Poaceae | Cosmop. | G | *Cynodon dactylon* (L.) Pers. | 1 | 1 |
| 78 | Cyperaceae | Steno-Medit. | G | *Cyperus capitatus* Vand. | 1 | |
| 79 | Poaceae | Steno-Medit. | H | *Dactylis glomerata* subsp. *hispanica* (Roth) Nyman | | 1 |
| 80 | Poaceae | Paleotrop. | T | *Dactyloctenium aegyptium* (L.) Willd. | | 1 |
| 81 | Alismataceae | Atl. | I | *Damasonium bourgaei* Coss. | | 1 |
| 82 | Apiaceae | Paleotemp. | H | *Daucus carota* L. | 1 | 1 |
| 83 | Apiaceae | Cosmop. | H | *Daucus carota* L. subsp. *carota* | 1 | |
| 84 | Apiaceae | W-Medit. | H | *Daucus carota* L. subsp. *maritimus* (Lam.) Batt. | 1 | |
| 85 | Brassicaceae | W-Medit. | T | *Diplotaxis erucoides* (L.) DC. | | 1 |
| 86 | Dipsacaceae | Euri-Medit. | H | *Dipsacus fullonum* L. | 1 | |
| 87 | Asteraceae | Euri-Medit. | H | *Dittrichia viscosa* (L.) Greuter | 1 | 1 |
| 88 | Apiaceae | Euri-Medit. | H | *Echinophora spinosa* L. | 1 | |
| 89 | Boraginaceae | Steno-Medit. | H | *Echium arenarium* Guss. | 1 | |
| 90 | Boraginaceae | Endem. Sic. | H | *Echium italicum* subsp. *siculum* (Lacaita) Greuter and Burdet | 1 | |
| 91 | Boraginaceae | Steno-Medit. | H | *Echium parviflorum* Moench | 1 | |
| 92 | Cyperaceae | Subcosmop. | G | *Eleocharis palustris* (L.) Roem. and Schult. | 1 | |
| 93 | Geraniaceae | Subcosmop. | T | *Erodium cicutarium* (L.) L'Hér. | | 1 |
| 94 | Geraniaceae | Steno-Medit. | T | *Erodium laciniatum* (Cav.) Willd. | 1 | |
| 95 | Apiaceae | Euri-Medit. | H | *Eryngium campestre* L. | | 1 |
| 96 | Apiaceae | SW-Medit. | H | *Eryngium dicothomum* Desf. | | 1 |
| 97 | Apiaceae | Medit.-Atl. | G | *Eryngium maritimum* L. | 1 | |
| 98 | Euphorbiaceae | Cosmop. | T | *Euphorbia helioscopia* L. | | 1 |
| 99 | Euphorbiaceae | Medit. | G | *Euphorbia hirsuta* L. | 1 | |
| 100 | Euphorbiaceae | Cosmop. | T | *Euphorbia peplus* L. | | 1 |
| 101 | Euphorbiaceae | W-Medit. | H | *Euphorbia segetalis* L. | | 1 |
| 102 | Euphorbiaceae | Steno-Medit. | H | *Euphorbia terracina* L. | 1 | |
| 103 | Apiaceae | Euri-Medit. | H | *Ferula communis* L. | | 1 |
| 104 | Poaceae | Subcosmop. | T | *Festuca danthonii* Asch. and Graebn. | 1 | 1 |
| 105 | Poaceae | Subcosmop. | T | *Festuca myuros* L. | | 1 |
| 106 | Poaceae | Medit.-Atl. | T | *Festuca fasciculata* Forssk. | 1 | |
| 107 | Moraceae | Medit.-Turan. | P | *Ficus carica* L. | 1 | 1 |
| 108 | Asteraceae | Steno-Medit. | T | *Filago pygmaea* L. | | 1 |
| 109 | Frankeniaceae | Steno-Medit. | Ch | *Frankenia hirsuta* L. | | 1 |
| 110 | Frankeniaceae | Steno-Medit. | T | *Frankenia pulverulenta* L. | | 1 |
| 111 | Asteraceae | Steno-Medit. | H | *Galactites tomentosus* Moench | 1 | 1 |
| 112 | Rubiaceae | Eurasiat. | T | *Galium aparine* L. | 1 | |
| 113 | Geraniaceae | Cosmop. | T | *Geranium dissectum* L. | | 1 |
| 114 | Geraniaceae | Paleotemp. | T | *Geranium rotundifolium* L. | 1 | 1 |
| 115 | Iridaceae | Euri-Medit. | G | *Gladiolus italicus* Mill. | 1 | 1 |

**Table A1.** *Cont.*

| N. | Family | Corology | Life Form | Species | Saline Priolo | Penisola Magnisi |
|---|---|---|---|---|---|---|
| 116 | Papaveraceae | Euri-Medit. | H | *Glaucium flavum* Crantz | | 1 |
| 117 | Asteraceae | Steno-Medit. | T | *Glebionis coronaria* (L.) Spach | | |
| 118 | Amaranthaceae | Circumbor. | Ch | *Halimione portulacoides* (L.) Aellen | 1 | |
| 119 | Asteraceae | Steno-Medit. | T | *Hedypnois rhagadioloides* (L.) F.W. Schmidt | 1 | 1 |
| 120 | Asteraceae | Euri-Medit. | T | *Helminthotheca echioides* (L.) Holub | 1 | |
| 121 | Brassicaceae | Subatl. | H | *Hirschfeldia incana* (L.) Lagr.-Foss. | 1 | 1 |
| 122 | Poaceae | Circumbor. | H | *Holcus lanatus* L. | 1 | |
| 123 | Poaceae | Euri-Medit. | T | *Hordeum murinum* subsp. *leporinum* (Link) Arcang. | 1 | 1 |
| 124 | Asteraceae | Steno-Medit. | H | *Hyoseris radiata* L. | | 1 |
| 125 | Poaceae | Paleotrop. | H | *Hyparrhenia hirta* (L.) Stapf | | 1 |
| 126 | Hypericaceae | Steno-Medit. | H | *Hypericum triquetrifolium* Turra | | 1 |
| 127 | Asteraceae | Steno-Medit. | T | *Hypochaeris achyrophorus* L. | 1 | 1 |
| 128 | Cyperaceae | Subcosmop. | H | *Isolepis cernua* (Vahl) Roem. and Schult. | 1 | |
| 129 | Juncaceae | Euri-Medit. | H | *Juncus acutus* L. | 1 | 1 |
| 130 | Juncaceae | Circumbor. | G | *Juncus articulatus* L. | 1 | |
| 131 | Juncaceae | Cosmop. | T | *Juncus bufonius* L. | | 1 |
| 132 | Juncaceae | Paleosubtrop. | G | *Juncus fontanesii* J.Gay | 1 | |
| 133 | Juncaceae | Euri-Medit. | T | *Juncus hybridus* Brot. | 1 | 1 |
| 134 | Juncaceae | Subcosmop. | G | *Juncus maritimus* Lam. | 1 | |
| 135 | Juncaceae | S-Medit. | G | *Juncus subulatus* Forssk. | 1 | |
| 136 | Plantaginaceae | Steno-Medit. | H | *Kickxia commutata* (Bernh. ex Rchb.) Fritsch | 1 | |
| 137 | Poaceae | Euri-Medit. | T | *Lagurus ovatus* L. | 1 | |
| 138 | Fabaceae | Euri-Medit. | T | *Lathyrus annuus* L. | 1 | |
| 139 | Fabaceae | Euri-Medit. | T | *Lathyrus aphaca* L. | 1 | 1 |
| 140 | Fabaceae | Euri-Medit. | T | *Lathyrus cicera* L. | | 1 |
| 141 | Fabaceae | Steno-Medit. | T | *Lathyrus clymenum* L. | | 1 |
| 142 | Fabaceae | Steno-Medit. | T | *Lathyrus ochrus* (L.) DC. | 1 | 1 |
| 143 | Malvaceae | Steno-Medit. | H | *Malva arborea* (L.) Webb and Berthel. | | |
| 144 | Brassicaceae | Euri-Medit. | T | *Lepidium coronopus* (L.) Al-Shehbaz | | 1 |
| 145 | Asteraceae | Medit.Atl. | Ch | *Limbarda crithmoides* (L.) Dumort. | 1 | 1 |
| 146 | Plumbacinaceae | Euri-Medit. | H | *Limonium narbonense* Mill. | 1 | |
| 147 | Plumbaginaceae | Endem. Sic. | Ch | *Limonium syracusanum* Brullo | 1 | 1 |
| 148 | Plantaginaceae | W-Medit. | T | *Linaria triphylla* (L.) Mill. | | 1 |
| 149 | Linaceae | Euri-Medit. | H | *Linum usitatissimum* subsp. *angustifolium* (Huds.) Thell. | | 1 |
| 150 | Brassicaceae | Steno-Medit. | Ch | *Lobularia maritima* (L.) Desv. | 1 | 1 |
| 151 | Poaceae | Paleotemp. | H | *Lolium arundinaceum* (Schreb.) Darbysh. | 1 | |
| 152 | Fabaceae | Euri-Medit. | H | *Lotus corniculatus* L. subsp. *preslii* (Ten.) P.Fourn. | 1 | |
| 153 | Fabaceae | Steno-Medit. | Ch | *Lotus cytisoides* L. | 1 | 1 |
| 154 | Fabaceae | Steno-Medit. | T | *Lotus edulis* L. | | 1 |
| 155 | Fabaceae | Steno-Medit. | T | *Lotus ornithopodioides* L. | | 1 |
| 156 | Primulaceae | Cosmop. | T | *Lysimachia arvensis* (L.) U. Manns and Anderb. | 1 | 1 |
| 157 | Lythraceae | Subcosmop. | T | *Lythrum hyssopifolia* L. | | 1 |
| 158 | Lythraceae | Steno-Medit. | H | *Lythrum junceum* Banks and Sol. | 1 | |
| 159 | Malvaceae | Subcosmop. | H | *Malva sylvestris* L. | 1 | |
| 160 | Solanaceae | Steno-Medit. | H | *Mandragora autumnalis* Bertol. | | 1 |
| 161 | Lamiaceae | Euri-Medit. | H | *Marrubium vulgare* L. | 1 | 1 |
| 162 | Brassicaceae | Steno-Medit. | T | *Matthiola tricuspidata* (L.) W.T. Aiton | 1 | 1 |
| 163 | Fabaceae | Euri-Medit. | T | *Medicago littoralis* Rohde ex Loisel. | 1 | 1 |
| 164 | Fabaceae | Euri-Medit. | Ch | *Medicago marina* L. | 1 | |

**Table A1.** *Cont.*

| N. | Family | Corology | Life Form | Species | Saline Priolo | Penisola Magnisi |
|---|---|---|---|---|---|---|
| 165 | Fabaceae | Euri-Medit. | T | *Medicago minima* (L.) L. | 1 | 1 |
| 166 | Fabaceae | Euri-Medit. | T | *Medicago polymorpha* L. | | 1 |
| 167 | Fabaceae | Steno-Medit. | T | *Medicago truncatula* Gaertn. | | 1 |
| 168 | Lamiaceae | Euri-Medit. | H | *Mentha pulegium* L. | 1 | 1 |
| 169 | Euphorbiaceae | Paleotemp. | T | *Mercurialis annua* L. | 1 | 1 |
| 170 | Aizoaceae | S-Medit. | T | *Mesembryanthemum nodiflorum* L. | | 1 |
| 171 | Lamiaceae | Endem. Ital. | Ch | *Micromeria graeca* subsp. *tenuifolia* (Ten.) Nyman | 1 | 1 |
| 172 | Lamiaceae | S-Medit. | Ch | *Micromeria nervosa* (Desf.) Benth. | | 1 |
| 173 | Plantaginaceae | Euri-Medit. | T | *Misopates orontium* (L.) Raf. | 1 | |
| 174 | Iridaceae | Steno-Medit. | G | *Moraea sisyrinchium* (L.) Ker Gawl. | | 1 |
| 175 | Asparagaceae | Euri-Medit. | G | *Muscari comosum* (L.) Mill. | | 1 |
| 176 | Asparagaceae | Steno-Medit. | G | *Muscari parviflorum* Desf. | | 1 |
| 177 | Myrtaceae | Steno-Medit. | P | *Myrtus communis* L. | 1 | |
| 178 | Ranunculaceae | Euri-Medit. | T | *Nigella damascena* L. | | 1 |
| 179 | Asteraceae | Steno-Medit. | T | *Notobasis syriaca* (L.) Cass. | | 1 |
| 180 | Apiaceae | Steno-Medit. | H | *Oenanthe globulosa* L. | 1 | |
| 181 | Oleaceae | Steno-Medit. | P | *Olea europaea* L. var. *sylvestris* (Mill.) Lehr | 1 | 1 |
| 182 | Poaceae | Medit.-Turan. | H | *Oloptum miliaceum* (L.) Röser and H.R. Hamasha | 1 | |
| 183 | Fabaceae | Steno-Medit. | T | *Onobrychis caput-galli* (L.) Lam. | 1 | 1 |
| 184 | Fabaceae | Euri-Medit. | H | *Ononis natrix* subsp. *ramosissima* (Desf.) Batt. | 1 | 1 |
| 185 | Fabaceae | Medit.-Turan. | T | *Ononis reclinata* L. | 1 | 1 |
| 186 | Fabaceae | Steno-Medit. | T | *Ononis variegata* L. | 1 | |
| 187 | Asteraceae | Steno-Medit. | H | *Onopordum illyricum* L. | | 1 |
| 188 | Orchidaceae | Steno-Medit | G | *Ophrys bertolonii* Moretti | 1 | |
| 189 | Cactaceae | Americ. | P | *Opuntia dillenii* (Ker Gawl.) Haw. | 1 | |
| 190 | Cactaceae | Neotrop. | P | *Opuntia ficus-indica* (L.) Mill. | | 1 |
| 191 | Asparagaceae | Steno-Medit. | G | *Ornithogalum gussonei* Ten. | | 1 |
| 192 | Oxalidaceae | Africana | G | *Oxalis pes-caprae* L. | 1 | 1 |
| 193 | Asteraceae | Euri-Medit. | H | *Pallenis spinosa* (L.) Cass. | | 1 |
| 194 | Amaryllidaceae | Steno-Medit. | G | *Pancratium maritimum* L. | 1 | |
| 195 | Poaceae | Euri-Medit. | T | *Parapholis cylindrica* (Willd.) Romero Zarco | | 1 |
| 196 | Poaceae | Medit.-Atl. | T | *Parapholis filiformis* (Roth) C.E. Hubb. | 1 | |
| 197 | Poaceae | Medit.-Atl. | T | *Parapholis incurva* (L.) C.E. Hubb. | 1 | 1 |
| 198 | Urticaceae | Euri-Medit. | H | *Parietaria judaica* L. | 1 | |
| 199 | Asteraceae | Steno-Medit. | Ch | *Phagnalon saxatile* (L.) Cass. | | 1 |
| 200 | Poaceae | Steno-Medit. | H | *Phalaris coerulescens* Desf. | 1 | |
| 201 | Poaceae | Subcosmop. | G | *Phragmites australis* (Cav.) Trin. ex Steud. | 1 | |
| 202 | Verbenaceae | Pantrop. | H | *Phyla nodiflora* (L.) Greene | 1 | |
| 203 | Anacardiaceae | S-Medit. | P | *Pistacia lentiscus* L. | | 1 |
| 204 | Plantaginaceae | Euri-Medit. | H | *Plantago coronopus* L. | | 1 |
| 205 | Plantaginaceae | Steno-Medit. | T | *Plantago lagopus* L. | 1 | 1 |
| 206 | Plantaginaceae | Steno-Medit. | H | *Plantago macrorhiza* Poir. | | 1 |
| 207 | Plantaginaceae | Eurasiat. | H | *Plantago media* L. | 1 | |
| 208 | Plantaginaceae | Steno-Medit. | H | *Plantago serraria* L. | | 1 |
| 209 | Poaceae | Euri-Medit. | T | *Poa infirma* Kunth | 1 | 1 |
| 210 | Poaceae | Paleosubtrop. | T | *Polypogon monspeliensis* (L.) Desf. | 1 | |
| 211 | Rosaceae | Paleotemp. | H | *Potentilla reptans* L. | 1 | |
| 212 | Rosaceae | Steno-Medit. | NP | *Poterium spinosum* L. | | 1 |
| 213 | Asparagaceae | Steno-Medit. | G | *Prospero autumnale* (L.) Speta | | 1 |
| 214 | Asteraceae | Steno-Medit. | T | *Pulicaria sicula* (L.) Moris | 1 | |
| 215 | Asteraceae | Steno-Medit. | H | *Pulicaria dysenterica* (L.) Bernh. | 1 | |

**Table A1.** *Cont.*

| N. | Family | Corology | Life Form | Species | Saline Priolo | Penisola Magnisi |
|----|--------|----------|-----------|---------|---------------|------------------|
| 216 | Rosaceae | Eurasiat. | P | *Pyrus spinosa* Forssk. | | 1 |
| 217 | Ranunculaceae | W-Medit. | T | *Ranunculus trilobus* Desf. | 1 | |
| 218 | Asteraceae | Steno-Medit. | H | *Reichardia picroides* (L.) Roth | | 1 |
| 219 | Resedaceae | Steno-Medit. | H | *Reseda alba* L. | 1 | |
| 220 | Rhamnaceae | Steno-Medit. | P | *Rhamnus alaternus* L. | 1 | 1 |
| 221 | Iridaceae | Steno-Medit. | G | *Romulea ramiflora* Ten. | | 1 |
| 222 | Rosaceae | Steno-Medit. | NP | *Rosa sempervirens* L. | 1 | |
| 223 | Poaceae | Paleotemp. | T | *Rostraria cristata* (L.) Tzvelev | | 1 |
| 224 | Rubiaceae | Steno-Medit. | P | *Rubia peregrina* L. | 1 | |
| 225 | Rosaceae | Euri-Medit. | NP | *Rubus ulmifolius* Schott | 1 | |
| 226 | Polygonaceae | Euri-Medit. | H | *Rumex pulcher* L. | 1 | 1 |
| 227 | Polygonaceae | Eurasiat. | H | *Rumex conglomeratus* Murray | | |
| 228 | Polygonaceae | Subcosmop. | H | *Rumex crispus* L. | 1 | |
| 229 | Polygonaceae | W-Medit. | H | *Rumex thyrsoides* Desf. | | 1 |
| 230 | Ruppiaceae | Cosmop. | I | *Ruppia maritima* L. | 1 | |
| 231 | Amaranthaceae | Euri-Medit. | Ch | *Salicornia fruticosa* (L.) L. | | |
| 232 | Amaranthaceae | W-Europ. | T | *Salicornia perennans* Willd. | 1 | |
| 233 | Amaranthaceae | Steno-Medit. | Ch | *Salicornia perennis* subsp. *alpini* (Lag.) Castrov. | 1 | |
| 234 | Salicaceae | Steno-Medit. | P | *Salix pedicellata* Desf. | 1 | |
| 235 | Amaranthaceae | Paleotemp. | T | *Salsola tragus* L. | 1 | |
| 236 | Lamiaceae | Euri-Medit. | H | *Salvia verbenaca* L. | | 1 |
| 237 | Gentianaceae | Euri-Medit. | T | *Schenkia spicata* (L.) G. Mans. | 1 | |
| 238 | Cyperaceae | Euri-Medit. | G | *Scirpoides holoschoenus* (L.) Soják | 1 | |
| 239 | Asteraceae | SW-Medit. | H | *Scolymus grandiflorus* Desf. | | 1 |
| 240 | Asteraceae | Euri-Medit. | H | *Scolymus hispanicus* L. | 1 | |
| 241 | Fabaceae | S-Medit. | T | *Scorpiurus vermiculatus* L. | | 1 |
| 242 | Crassulaceae | SW-Medit. | T | *Sedum caeruleum* L. | 1 | 1 |
| 243 | Asteraceae | Cosmop. | T | *Senecio vulgaris* L. | 1 | |
| 244 | Caryophyllaceae | S-Medit. | T | *Silene bellidifolia* Jacq. | | 1 |
| 245 | Caryophyllaceae | Steno-Medit. | T | *Silene colorata* Poir. | 1 | 1 |
| 246 | Caryophyllaceae | Steno-Medit. | T | *Silene niceensis* All. | 1 | |
| 247 | Caryophyllaceae | Steno-Medit. | T | *Silene sedoides* Poir. | | 1 |
| 248 | Caryophyllaceae | Paleotemp. | H | *Silene vulgaris* (Moench) Garcke | | 1 |
| 249 | Asteraceae | Medit.-Turan. | H | *Silybum marianum* (L.) Gaertn. | 1 | 1 |
| 250 | Brassicaceae | E-Medit. | T | *Sinapis alba* L. | 1 | |
| 251 | Dipsacaceae | Steno-Medit. | H | *Sixalix atropurpurea* (L.) Greuter and Burdet | 1 | 1 |
| 252 | Apiaceae | Medit.-Atl. | H | *Smyrnium olusatrum* L. | 1 | |
| 253 | Asteraceae | Eurasiat. | H | *Sonchus asper* (L.) Hill | | 1 |
| 254 | Asteraceae | Steno-Medit. | G | *Sonchus bulbosus* (L.) Kilian and Greuter | 1 | |
| 255 | Asteraceae | Cosmop. | H | *Sonchus oleraceus* L. | 1 | 1 |
| 256 | Asteraceae | Steno-Medit. | H | *Sonchus tenerrimus* L. | | 1 |
| 257 | Caryophyllaceae | Subcosmop. | T | *Spergularia marina* (L.) Besser | 1 | 1 |
| 258 | Poaceae | Subtrop. | G | *Sporobolus virginicus* (L.) Kunth | 1 | |
| 259 | Lamiaceae | Steno-Medit. | Ch | *Stachys major* (L.) Bartolucci and Peruzzi | | 1 |
| 260 | Lamiaceae | Steno-Medit. | T | *Stachys romana* (L.) E.H.L. Krause | 1 | 1 |
| 261 | Poaceae | Steno-Medit. | T | *Stipellula capensis* (Thunb.) Röser and H.R. Hamasha | 1 | 1 |
| 262 | Amaranthaceae | Cosmop. | T | *Suaeda maritima* (L.) Dumort. | 1 | |
| 263 | Amaranthaceae | Cosmop. | NP | *Suaeda vera* J. F. Gmelin | 1 | 1 |
| 264 | Asteraceae | Neotrop. | H | *Symphyotrichum squamatum* (Spreng.) G. L. Nesom | 1 | |
| 265 | Tamaricaceae | W-Medit. | P | *Tamarix africana* Poir. | 1 | 1 |
| 266 | Tamaricaceae | S-Medit. | P | *Tamarix arborea* (Ehrenb.) Bunge | 1 | |

**Table A1.** *Cont.*

| N. | Family | Corology | Life Form | Species | Saline Priolo | Penisola Magnisi |
|---|---|---|---|---|---|---|
| 267 | Tamaricaceae | W-Medit. | P | *Tamarix gallica* L. | 1 | |
| 268 | Lamiaceae | Europ.-Caucas. | H | *Teucrium scordium* L. | 1 | |
| 269 | Apiaceae | S-Medit. | H | *Thapsia garganica* L. | | 1 |
| 270 | Poaceae | Submedit. | H | *Thinopyrum flaccidifolium* (Boiss. and Heldr.) Moustakas | 1 | |
| 271 | Poaceae | Euri-Medit. | G | *Thinopyrum junceum* (L.) Á. Löve | 1 | |
| 272 | Lamiaceae | Steno-Medit. | Ch | *Thymbra capitata* (L.) Cav. | | 1 |
| 273 | Thymelaeaceae | S-Medit. | Ch | *Thymelaea hirsuta* (L.) Endl. | 1 | |
| 274 | Apiaceae | Steno-Medit. | T | *Tordylium apulum* L. | | 1 |
| 275 | Apiaceae | Subcosmop. | T | *Torilis arvensis* (Huds.) Link | 1 | |
| 276 | Asteraceae | Euri-Medit. | H | *Tragopogon porrifolius* L. | | 1 |
| 277 | Fabaceae | Paleotemp. | T | *Trifolium campestre* Schreb. | 1 | 1 |
| 278 | Fabaceae | Euri-Medit. | T | *Trifolium cherleri* L. | 1 | 1 |
| 279 | Fabaceae | Paleotemp. | H | *Trifolium fragiferum* L. | 1 | |
| 280 | Fabaceae | Euri-Medit. | T | *Trifolium nigrescens* Viv. | 1 | 1 |
| 281 | Fabaceae | Paleotemp. | H | *Trifolium resupinatum* L. | | 1 |
| 282 | Fabaceae | Euri-Medit. | T | *Trifolium scabrum* L. | 1 | 1 |
| 283 | Fabaceae | Euri-Medit. | T | *Trifolium stellatum* L. | | 1 |
| 284 | Fabaceae | Paleotemp. | T | *Trifolium tomentosum* L. | | 1 |
| 285 | Fabaceae | S-Medit. | T | *Trigonella sulcata* (Desf.) Coulot and Rabaute | | 1 |
| 286 | Poaceae | Steno-Medit. | T | *Trisetaria aurea* (Ten.) Banfi and Galasso | | 1 |
| 287 | Poaceae | Medit.-Turan. | T | *Triticum vagans* (Jord. and Fourr.) Greuter | | 1 |
| 288 | Typhaceae | Pantrop. | G | *Typha domingensis* (Pers.) Steud. | 1 | |
| 289 | Typhaceae | Circumbor. | G | *Typha angustifolia* L. | 1 | |
| 290 | Ulmaceae | Europ.-Caucas. | P | *Ulmus minor* Mill. | 1 | |
| 291 | Asteraceae | Euri-Medit. | T | *Urospermum picroides* (L.) Scop. ex F.W. Schmidt | 1 | 1 |
| 292 | Urticaceae | S-Medit. | T | *Urtica membranacea* Poir. | 1 | |
| 293 | Asteraceae | Neotrop. | T | *Symphyotrichum squamatum* (Spreng.) G.L. Nesom | | |
| 294 | Rubiaceae | Steno-Medit. | T | *Valantia muralis* L. | | 1 |
| 295 | Scrophulariaceae | Euri-Medit. | H | *Verbascum sinuatum* L. | 1 | 1 |
| 296 | Plantaginaceae | Cosmop. | H | *Veronica anagallis-aquatica* L. | 1 | |
| 297 | Fabaceae | Euri-Medit. | T | *Vicia hybrida* L. | | 1 |
| 298 | Fabaceae | S-Europ. | T | *Vicia melanops* Sm. | 1 | 1 |
| 299 | Fabaceae | Steno-Medit. | T | *Vicia sativa* L. | | 1 |
| 300 | Fabaceae | Steno-Medit. | H | *Vicia villosa* Roth | 1 | 1 |
| 301 | Fabaceae | Euri-Medit. | T | *Ervum gracile* DC. (=Vicia tenuissima (Bieb.) Sch. and Th.) | 1 | 1 |
| 302 | Lamiaceae | Medit.-Turan. | P | *Vitex agnus-castus* L. | 1 | |
| 303 | Asteraceae | S-Europ. | T | *Xanthium italicum* Moretti | 1 | |
| 304 | Rhamnaceae | S-Medit. | P | *Ziziphus lotus* (L.) Lam. | | 1 |

## Appendix B. Syntaxonomical Scheme of the Vegetation Units Recorded from the "Saline di Priolo" SAC (SE Sicily)

CAKILETEA MARITIMAE Tüxen and Preising ex Br.-Bl. and Tüxen 1952
EUPHORBIETALIA PEPLIS Tüxen 1950
EUPHORBION PEPLIS Tüxen 1950
*1. Salsolo-Cakiletum maritimae* Costa and Mansanet 1981 corr. Rivas-Martínez et al., 1992
EUPHORBIO PARALIAE-AMMOPHILETEA AUSTRALIS Géhu and Rivas-Martínez in Rivas Martínez, Asensi, Díez-Garretas, Molero, Valle, Cano, Costa and Díaz 2011

AMMOPHILETALIA AUSTRALIS Br.-Bl. 1933

AMMOPHILION AUSTRALIS Br.-Bl. 1921 corr. Rivas-Martìnez, Costa Izco in Rivas-Martìnez, Lousa, T.

E.Diaz, Fernandez-Gonzalez and J.C.Costa 1990

*2. Cypero capitati-Agropyretum juncei* Kühnholtz-Lordat (1923) Br.-Bl. 1933

ONONIDION RAMOSISSIMAE Pignatti 1952

*3. Centaureo sphaerocephalae-Ononidetum ramosissimae* Br.-Bl. e Frei in Frei, 193

HELIANTHEMETEA GUTTATI (Br.-Bl. in Br.-Bl., Roussine and Nègre 1952) Rivas Goday and Rivas-Martìnez 1963 em. Rivas-Martìnez 1978

CUTANDIETALIA MARITIMAE Rivas- Martìnez, Díez-Garretas, and Asensi 2002

ALKANNO-MARESION NANAE Rivas Goday ex Rivas Goday and Rivas-Martínez 1963 corr. Diaz-Garretas et al., 2001

*4. Sileno coloratae-Ononidetum variegatae* Gèhu and Gèhu-Franck 1986

RUPPIETEA MARITIMAE Tuxen ex Den Hartog and Segal 1964

RUPPIETALIA MARITIMAE Tuxen ex Den Hartog and Segal 1964

RUPPION MARITIMAE Br.-Bl. ex Br.-Bl., Roussine and Nègre 1952

*5. Enteromorpho intestinalidis-Ruppietum maritimae* Westhoff ex R.Tx. and Böckelmann 1957

PHRAGMITO-MAGNOCARICETEA Klika in Klika and Novák 1941

PHRAGMITETALIA Koch 1926

PHRAGMITION Koch 1926

*6. Phragmitetum communis* (Koch 1926) Schmale 1939

*7. Typhetum domingensis* Brullo, Minissale and Spamp. 1994

NASTURTIO-GLYCERIETALIA Pign. 1954

GLYCERIO-SPARGANION Br.-Bl. and Sissing in Boer 1942

*8. Eleocharido-Alismetum lanceolati* Minissale and Spampinato 1987

*9. Bolboschoeno maritimi-Alismetum lanceolati* ass. nov. hoc loco

MOLINIO-ARRHENATHERETEA R.Tx.1937

PASPALO-AGROSTION SEMIVERTICILLATI Br.-Bl. in Br.-Bl. Roussine and Negre 1952

PASPALO-HELEOCHLOETALIA Br.-Bl. ex Rivas Goday 1956

*10. Euphorbio hirsutae-Lotetum preslii* ass. nov. hoc loco

SALICORNIETEA FRUTICOSAE Br.-Bl. et Tx. ex A. Bolòs y Vayreda et. O. de Bolòs in A. Bolòs et Vayreda 1950

SALICORNIETALIA FRUTICOSAE Br.-Bl. 1933

SALICORNION FRUTICOSAE Br.-Bl. 1933

*11. Junco subulati-Sarcocornietum alpini* Brullo et Sciandrello in Giusso et al., 2008

ARTHROCNEMION GLAUCI Rivas-Mart. et Costa M. 1984

*12. Arthrocaulo meridionalis-Juncetum subulati* Brullo et Furnari 1976 *nom. corr.* Sciandrello et al., 2019

*13. Limonio virgati-Arthrocnemetum macrostachyi* Biondi, Casavecchia and Guerra 2006

SUAEDION BREVIFOLIAE Br.-Bl. et O. de Bolòs 1958 (= *Suaedion verae* Brullo et Furnari 1988)

*14. Halimiono-Suaedetum verae* Molinier et Tallon 1970 corr. Géhu 1984

INULION CRITHMOIDIS Brullo et Furnari 1988

*15. Agropyro scirpei-Inuletum crithmoidis* Brullo in Brullo et al., 1988

THERO-SUAEDETEA SPLENDENTIS Rivas-Martínez 1972

THERO-SALICORNIETALIA Tüxen in Tüxen et Oberdorfer ex Géhu et Géhu-Franck 1984

Salicornion patulae Géhu et Géhu-Franck ex Rivas-Martínez 1990

*16. Suaedo-Salicornietum patulae* Brullo et Furnari ex Géhu et Géhu-Franck 1984

JUNCETEA MARITIMI Br.Bl. in Br.-Bl., Roussine and Nègre 1952

JUNCETALIA MARITIMI Br.Bl. ex Horvatic 1934

JUNCION MARITIMI Br.Bl. ex Horvatic 1934

*17. Juncetum maritimo-acuti* Horvatic 1934 (*Juncus acutus* comm.)

HALO-ARTEMISION COERULESCENTIS Pignatti 1953

*18. Elymetum atherici* Pellizzari, Merloni et Piccoli 1998 (=*Thinopyrum acutum* (DC.) Banfi)

SAGINETEA MARITIMAE Westhoff, Van Leeuwen et Adriani 1962

SAGINETALIA MARITIMAE Westhoff, Van Leeuwen et Adriani 1962

SILENO SEDOIDIS-CATAPODION BALEARICI de Foucault and Bioret 2010 corr. Tomaselli et al., 2020

*19. Parapholido incurvae-Spergularietum marinae* ass. nov. hoc loco

ISOËTO-NANOJUNCETEA Br.-Bl. and R. Tx. ex Westhoff, Dijk and Passchier 1946

Isoëtetalia Br.-Bl. 1936

Isoëtion Br.-Bl. 1936
*20. Lythro hyssopifoliae-Crassuletum vaillantii* Bagella et al., 2009
NERIO-TAMARICETEA Br.-Bl. et O.Bolòs 1958
TAMARICETALIA AFRICANAE Br.-Bl. et O. Bolòs 1958
TAMARICION AFRICANAE Br.-Bl. et O.Bolòs 1958
*21. Inulo crithmoidis-Tamaricetum africanae* Gamisans 1992 (=*Limbardo crithmoidis-Tamaricetum africanae* Sciandrello et al., 2019)
CRITHMO-LIMONIETEA Br.-Bl. in Br- Bl., Roussine and Nègre1952
CRITHMO-LIMONIETALIA Molinier 1934
CRITHMO-LIMONION MOLINIER 1934
*22. Limonietum syracusani* Bartolo, Brullo and Marcenò 1982
PEGANO HARMALAE-SALSOLETEA VERMICULATAE Br-Bl and O.Bolòs 1958
SALSOLO VERMICULATAE-PEGANETALIA HARMALAE Br.-Bl. and O. Bolòs 1954
Artemision arborescentis Géhu et al., 1986
*23. Atriplici halimi-Artemisietum arborescentis* Biondi 1988 (*Artemisia arborescens* comm.)
CISTO-MICROMERIETEA Oberd. 1954
*CISTO-ERICETALIA Horvatic 1958*
*CISTO-ERICION Horvatic 1958*
*24. Thymbra capitata* comm.
LYGEO SPARTI-STIPETEA TENACISSIMAE Rivas-Martinez 1978
CYMBOPOGONO-BRACHYPODIETALIA RAMOSI Horvatić 1963
*HYPARRHENENION HIRTAE* Brullo, Minissale and Spamp. in C. Brullo et al., 2010
*25. Hyparrhenietum hirto-pubescentis* A.and O. Bolòs and Br.-Bl. in A.and O. Bolòs 1950
BROMO-ORYZOPSION MILIACEAE O.Bolòs 1970
*26. Oloptum miliaceum* comm.
STIPO-TRACHYNIETEA DISTACHYAE Brullo in Brullo, Scelsi and Spampinato 2001
TRACHYNIETALIA DISTACHYAE Rivas-Martínez 1978
TRACHYNION DISTACHYAE Rivas-Martínez 1978
*27. Thero-Sedetum caerulei* Brullo 1975
CHENOPODIETEA Br.-Bl. in Br.-Bl. et al., 1952
BROMETALIA RUBENTI-TECTORUM (Rivas Goday et Rivas-Martinez 1973) Rivas-Martinez and Izco 1977
ECHIO PLANTAGINEI-GALACTITION TOMENTOSAE O. Bolòs and Molinier 1969
*28. Stipellula capensis* comm.

## Appendix C. Localities and Dates of Phytosociological Relevés

Table 1: *Parapholido incurvae-Spergularietum marinae* ass. nova (Rel. 17–22, Scogliera Penisola Magnisi, 30.03.2021, Cambria, Minissale, Sciandrello, Rel. 23–27 Scogliera Penisola Magnisi, 09.04.2021 (Cambria, Minissale, Sciandrello, Tavilla).

Table 2: *Bolboschoeno-Alismetum lanceolati* ass. nova (Rel. 22–24, 26–27, 30–32, pozze Saline di Priolo, 21.04.2021, Cambria, Minissale, Ranno, Sciandrello, Tavilla)

Table 3: *Euphorbio hirsutae-Lotetum preslii* ass. nova (Rel. 40–41, Saline Priolo, pozza lunga 21.09.2021, Sciandrello).

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
