# Peer review of "The Role of Vegetation Monitoring in the Conservation of Coastal Habitats N2000: A Case Study of a Wetland Area in Southeast Sicily (Italy)"

_land, doi:10.3390/land13010062_

Round 1
Reviewer 1 Report
Comments and Suggestions for Authors
1. Be careful while numbering the subheadings. Many subheadings are given the same numbering.
2. The objectives of study have been repeated in the introduction.
3. Is it necessary to write Thapos in italics (L.52)?
4. Better to write the forty square kilometers as 40 km2 (L-56)
5. Provide suitable subheadings under material and methods, and a figure/map for the study area.
6. L-83. The authors have used ‘such as’ and ‘among others’ simultaneously in a sentence. Please edit
7. L-93. Edit ‘arount’
8. L.94. Add full stop after [34]
9. I suggest the authors to add a study area map
10. Make sure all given binomials names are in italics (L. 230-232)
11. It is not the right way to mention the figure number with the subheadings.
12. Unnecessary punctuation marks have been noticed throughout the manuscript (eg: L. 251-253)
13. L-276-278. Some references are not provided with reference numbers. Please check it and do not use both references and numbers together too. Try to avoid cited references throughout the text to keep a uniform pattern.
14. Table 1 doesn't seem aesthetically pleasing and is confusing. Similarly, table 2&3.
15. L-296-307. Check all the cited references and provide reference numbers
16. Provide a high resolution image for figure 2,5,6,8,9 and 10
17. Please give the figure number and details under the figure (Ref.fig 6).
18. In table 6, some unnecessary green markings have been noticed in first column and the heading of first column doesn't begin with the capital letter. The table is placed amidst the table heading too.
19. No need to repeat entire scientific name if the genus part is same for multiple species. The authors can use first letter of the genus name after its first appearance in the manuscript.
20. Please check the entire manuscript for grammatical errors and poor wordings.
21. Arrange the spaces of each column in the tables to avoid misplacing of the contents in the columns.
22. Please keep supplementary files after references
Comments on the Quality of English LanguageMinor English editing required
Author Response
REV1
- Be careful while numbering the subheadings. Many subheadings are given the same numbering. Done
- The objectives of study have been repeated in the introduction. Done
- Is it necessary to write Thapos in italics (L.52)? Done
- Better to write the forty square kilometers as 40 km2 (L-56) Done
- Provide suitable subheadings under material and methods, and a figure/map for the study area. see answer 9
- L-83. The authors have used ‘such as’ and ‘among others’ simultaneously in a sentence. Please edit Done
- L-93. Edit ‘arount’ Done
- L.94. Add full stop after [34] Done
- I suggest the authors to add a study area map. We did not consider it appropriate to include a map of the study area so as not to burden the text with figures, also because the study area is indicated in figures no. 8,9,11. But if the reviewer deems it useful to include a map of the study area, we can include it.
- Make sure all given binomials names are in italics (L. 230-232) Done
- It is not the right way to mention the figure number with the subheadings. ?????
- Unnecessary punctuation marks have been noticed throughout the manuscript (eg: L. 251-253). Done
- L-276-278. Some references are not provided with reference numbers. Please check it and do not use both references and numbers together too. Try to avoid cited references throughout the text to keep a uniform pattern. Done
- Table 1 doesn't seem aesthetically pleasing and is confusing. Similarly, table 2&3. The tables in the text are editable, the journal can modify them as they wish.
- L-296-307. Check all the cited references and provide reference numbers Done
- Provide a high resolution image for figure 2,5,6,8,9 and 10 Done
- Please give the figure number and details under the figure (Ref.fig 6). Done
- In table 6, some unnecessary green markings have been noticed in first column and the heading of first column doesn't begin with the capital letter. The table is placed amidst the table heading too. Done
- No need to repeat entire scientific name if the genus part is same for multiple species. The authors can use first letter of the genus name after its first appearance in the manuscript. Done
- Please check the entire manuscript for grammatical errors and poor wordings. Done
- Arrange the spaces of each column in the tables to avoid misplacing of the contents in the columns. Done
- Please keep supplementary files after references Done
Reviewer 2 Report
Comments and Suggestions for Authors
The paper investigated the floristic composition, structure, conservation status, and trends of the habitat types. And three new plant associations were described, including two in the wetland and one on the rocky coasts. It also analyzes the spatiotemporal changes of the wetlands/coastal dune system and assess the loss/reduction of the habitats over the last 70 years.This topic is very interesting and the content is rich. I am not very familiar with vegetation observation and description.
I will evaluate and suggest changes in the spatiotemporal changes of the wetlands/coastal dune system.
1. The content in Figure 8 does not match the year of the Current vegetation map (2015), and the legend is not clear. Please add the latest land classification data and figures.
2. The total number of grids in Table 5 is different. Why?
3. What is the relationship between vegetation observation and temporal and spatial changes of coastal wetlands? Given the lack of historical data, can the early vegetation be related to the present?
4. In addition, what innovative conclusions can be drawn from changes in land vegetation cover?
Author Response
The paper investigated the floristic composition, structure, conservation status, and trends of the habitat types. And three new plant associations were described, including two in the wetland and one on the rocky coasts. It also analyzes the spatiotemporal changes of the wetlands/coastal dune system and assess the loss/reduction of the habitats over the last 70 years. This topic is very interesting and the content is rich. I am not very familiar with vegetation observation and description.
I will evaluate and suggest changes in the spatiotemporal changes of the wetlands/coastal dune system.
- The content in Figure 8 does not match the year of the Current vegetation map (2015), and the legend is not clear. Please add the latest land classification data and figures. Done
- The total number of grids in Table 5 is different. Why? thank you for this observation, we have re-done the calculations, in fact the difference between the two dates is around 14 ha, this differences is mainly linked to a strong process of erosion of the coastline in recent decades, as also highlighted in the text.
- What is the relationship between vegetation observation and temporal and spatial changes of coastal wetlands? Given the lack of historical data, can the early vegetation be related to the present? Due to the lack of historical phytosociological data it is not possible to relate the current floristic composition with that of the past, but thanks to the photos of historical areas it is possible to compare and quantify the surfaces of the habitats in the two periods and analyze the spatial changes. Es. the reduction of the dunes/wetland area, as well as the development of communities that were not present before, such as the Phragmites australis monophytic vegetation which arrived in recent decades due to the hyper nitrification of the land by surrounding agriculture, as well as weed vegetation on uncultivated land, etc. In conclusion, the vegetation of the past is partly correlated with the vegetation of the past but also modified from a floristic and in some cases also structural point of view.
- In addition, what innovative conclusions can be drawn from changes in land vegetation cover? Done
Reviewer 3 Report
Comments and Suggestions for Authors
Review Report
Article title: The role of vegetation monitoring in the conservation of coastal habitats N2000: a case study of a wetland area in southeast Sicily (Italy)
Journal: Land-2751749
Summary
The article focuses on the investigation of the floristic composition, structure, conservation status, and trends of the habitat types in the coastal wetland area in south-eastern Sicily. the topic deserves attention and the content and findings are good. However, the research suffers from some lacks tat have been commented in the following subsections;
Abstract and Keywords
- The goals of the study are supposed to be stated in the abstract
-The keywords are 7 that need to be reduced to 3-5
Introduction
- The introduction section lacks a compelling argument for the body of destructive policies and actions taken by humans that have negatively affected the wetland under study.
- The contributions of the study to the current literature have been missed
Methods
- The method of Euclidean coefficient and beta-flexible algorithm need to be explained and justified as a proper method for the case study
Results
- There is too much data and text in this section without a good organization which makes the article hard to follow. It is recommended to set research goals and questions first and then organize all data and analysis around them.
- Bad conservation status is different from the ways in which human pressure and activities lead to coastal wetlands of the Mediterranean vulnerability. However, it seems that these two different concepts have been mixed together. Therefore, it is highly recommended that these concepts are analyzed based on the findings and after illustrating the current situation of the case study.
- The article exceeds the word limit and it should be reduced to at most 10,000 in 25 pages
Comments on the Quality of English Language
Moderate editing of English language required
Author Response
REV3
I think the article suffers from two or third central questions that can organize their research structure. Therefore, the authors can clearly design these question and then reorganize their various sections around them.
- I think after reorganizing the article, as I mentioned in the review report, the authors can change the title so that cover all the content
- Methodologically speaking, the first paragraph of the material and method section is totally irrelevant as it has addressed the area specifications rather speaking about the methods. Moreover, I couldn't see any specific method as well as methodological approach proposing for analyzing the data. Using of maps in the article is weak while it should be the strength of the paper. We prefer to maintain our title and setting.
- Conclusion is too long to be scientific and it should be shortened significantly. Done
Summary: The article focuses on the investigation of the floristic composition, structure, conservation status, and trends of the habitat types in the coastal wetland area in south-eastern Sicily. the topic deserves attention and the content and findings are good. However, the research suffers from some lacks tat have been commented in the following subsections;
Abstract and Keywords: The goals of the study are supposed to be stated in the abstract; The keywords are 7 that need to be reduced to 3-5. Done
Introduction:
The introduction section lacks a compelling argument for the body of destructive policies and actions taken by humans that have negatively affected the wetland under study; This is not the objective of our paper.
- The contributions of the study to the current literature have been missed. Our study represents the first contribution on the flora and vegetation of the site.
Methods
- The method of Euclidean coefficient and beta-flexible algorithm need to be explained and justified as a proper method for the case study. Done
Results
- There is too much data and text in this section without a good organization which makes the article hard to follow. It is recommended to set research goals and questions first and then organize all data and analysis around them. Done
- Bad conservation status is different from the ways in which human pressure and activities lead to coastal wetlands of the Mediterranean vulnerability. However, it seems that these two different concepts have been mixed together. Therefore, it is highly recommended that these concepts are analyzed based on the findings and after illustrating the current situation of the case study. Our objective is the analysis of the conservation status of the communities/habitats according to our observations and field surveys, analyzing the structure and floristic composition of the communities in order to evaluate the conservation status. Also in line with the Italian interpretation Manual of the habitats [39].
- The article exceeds the word limit and it should be reduced to at most 10,000 in 25 pages. Please note that Tabs S1-S4 are supplementary. If the journal deems it a Floristic Appendix it can be moved as a supplement file (even if we prefer in text).
Reviewer 4 Report
Comments and Suggestions for Authors
The article presents valuable research results on changes in vegetation cover within valuable wetland habitats in southeastern Sicily. Classic research methods used in monitoring Natura 2000 habitats were used, and statistical analyzes were carried out. Changes were presented using vegetation maps, aerial photos and key landscape metrics. The publication is an important contribution to the protection of valuable and threatened ecosystems in Europe.
Author Response
REV4
The article presents valuable research results on changes in vegetation cover within valuable wetland habitats in southeastern Sicily. Classic research methods used in monitoring Natura 2000 habitats were used, and statistical analyzes were carried out. Changes were presented using vegetation maps, aerial photos and key landscape metrics. The publication is an important contribution to the protection of valuable and threatened ecosystems in Europe. OK, Tks
Round 2
Reviewer 2 Report
Comments and Suggestions for Authors
The manuscript has been revised and improved.
The content of the article is very detailed, and it can be carefully examined and considered for publication.
Reviewer 3 Report
Comments and Suggestions for Authors
The article has not been significantly improved as I expected based on the comments I provided in the previous referee round and I am afraid this work is not proper for publication in Land.
I think the article suffers from two or third central questions that can organize their research structure. Therefore, the authors can design these questions and then reorganize their various sections around them. Methodologically speaking, the first paragraph of the material and method section is irrelevant as it has addressed the area specifications rather than speaking about the methods. Moreover, I couldn't see any specific method as well as methodological approach proposed for analyzing the data. Using of maps in the article is weak while it should be the strength of the paper. The introduction section lacks a compelling argument for the body of destructive policies and actions taken by humans that have negatively affected the wetland under study; This is not the objective of our paper. The method of Euclidean coefficient and beta-flexible algorithm need to be explained and justified as a proper method for the case study. In sum, the work is more similar to a research report than a scientific article in the current format in the lack of sufficient improvements and as a result, I don't think it would be a good alternative to be published in Land. Comments on the Quality of English LanguageMinor editing of English language required